# Towards High-Order Mean Flow Generative Models: Feasibility, Expressivity, and Provably Efficient Criteria

## Abstract

Generative modelling has seen significant advances through simulation-free paradigms such as Flow Matching, and in particular, the MeanFlow framework, which replaces instantaneous velocity fields with average velocities to enable efficient single-step sampling. In this work, we introduce a theoretical study on second-order MeanFlow, a novel extension that incorporates average acceleration fields into the MeanFlow objective. We first establish the feasibility of our approach by proving that the average acceleration satisfies a generalized consistency condition analogous to first-order MeanFlow, thereby supporting stable, one-step sampling and tractable loss functions. We then characterize its expressivity via circuit complexity analysis, showing that under mild assumptions, the second-order MeanFlow sampling process can be implemented by uniform threshold circuits within the $\mathsf{TC}^0$ class. Finally, we derive provably efficient criteria for scalable implementation by leveraging fast approximate attention computations: we prove that attention operations within the second-order MeanFlow architecture can be approximated to within $1/\operatorname{poly}(n)$ error in time $n^{2+o(1)}$. Together, these results lay the theoretical foundation for high-order flow matching models that combine rich dynamics with practical sampling efficiency.

## 1 Introduction

Generative modeling has witnessed remarkable progress in recent years, driven by the development of flexible, simulation-free paradigms such as Flow Matching (FM) (Lipman et al., 2023; Albergo & Vanden-Eijnden, 2023; Liu et al., 2023b). By regressing vector fields between latent noise distributions and complex data distributions, flow matching provides an efficient alternative to continuous normalizing flows (Chen et al., 2018), achieving state-of-the-art performance in image and video generation (Frans et al., 2025; Polyak et al., 2024; Esser et al., 2024; Jin et al., 2025).

Recently, MeanFlow (Geng et al., 2025a) has emerged as a promising variant of flow matching. Unlike traditional FM models that predict instantaneous velocity fields, MeanFlow learns average velocities across time intervals and leverages Jacobian-vector product (JVP) computations for training. This not only simplifies optimization but also enables fast inference via a consistency condition that facilitates efficient single-step sampling (Song et al., 2023a; Song & Dhariwal, 2024). Given its practical advantages in training stability and inference speed, MeanFlow has become an important direction in modern generative modeling.

In this paper, we explore a fundamental theoretical question:

> *Can the MeanFlow framework be extended beyond first-order dynamics to incorporate second-order flow information such as acceleration fields?*

The motivation for this question is rooted in recent efforts in high-order flow matching (Gong et al., 2025b; Chen et al., 2025a;b; Liang et al., 2025), which demonstrate that capturing the influence of higher-order dynamics on the overall flow geometry (e.g., modeling both velocity and acceleration fields) enhances expressivity and improves generation quality (Chen, 2023; Hang & Gu, 2024; Lin et al., 2024). Inspired by this, we propose and study second-order MeanFlow, a novel framework

that generalizes average velocity to average acceleration, thereby creating a more expressive and consistent generative flow.

We provide a comprehensive theoretical analysis of second-order MeanFlow, addressing its feasibility, expressivity, and computational efficiency. Specifically, our contributions can be summarized as follows: **(i) Feasibility (Theorems 3.19, 3.22, and 3.23):** We introduce a formulation of average acceleration and show it satisfies a generalized consistency condition, enabling stable and fast sampling analogous to first-order MeanFlow. **(ii) Expressivity (Theorem 4.4):** We analyze the computational expressivity of second-order MeanFlow through the lens of circuit complexity theory, showing that the model can be simulated within the $\mathsf{TC}^0$ class under reasonable assumptions. **(iii) Provably Efficient Criteria (Theorem 5.8):** We prove that second-order MeanFlow can use fast approximation attention computation, to achieve $n^{2+o(1)}$ computation time complexity with less than $1/\operatorname{poly}(n)$ approximation error.

Together, our findings establish second-order MeanFlow as a theoretically grounded and practically feasible generalization of the MeanFlow family. We believe this work opens up new directions in the development of high-order flow models and their applications to fast and expressive generative modeling.

**Roadmap.** Section 2 provides the preliminaries. We then prove the feasibility of second-order MeanFlow in Section 3, analyze the circuit complexity of MeanFlow in Section 4, and provide a provable efficiency analysis in Section 5. We conclude our work in Section 6.

## 2 PRELIMINARY

In this section, we first introduce the notation used across this work. Then, we present the preliminary in flow matching. Next, we show the previous work of second-order flow matching.

### 2.1 NOTATION

We use $\Pr[]$ to denote the probability. We use $\mathbb{E}[]$ to denote the expectation. We use $\operatorname{Var}[]$ to represent the variance. Let $\|x\|_p$ be the $\ell_p$-norm of a vector $x \in \mathbb{R}^n$, i.e. $\|x\|_p := (\sum_{i=1}^{n} |x_i|^p)^{1/p}$. Specifically, we define $\ell_1$, $\ell_2$, and $\ell_\infty$ vector norm as $\|x\|_1 := \sum_{i=1}^{n} |x_i|$, $\|x\|_2 := (\sum_{i=1}^{n} x_i^2)^{1/2}$, and $\|x\|_\infty := \max_{i \in [n]} |x_i|$. We use $p(\cdot)$ to denote the probability density function for a distribution. Let $\operatorname{Uniform}[a, b]$ be a uniform distribution in the interval $[a, b]$. Let $\circ$ denote the Hadamard product. We employ the operator $\|$ to denote merging two matrices or vectors along the final dimension.

### 2.2 BASIC CONCEPTS IN FLOW MATCHING

Generative models like Flow Matching (Albergo & Vanden-Eijnden, 2023; Lipman et al., 2023; Liu et al., 2023b) are designed to learn the velocity fields that transform one probability distribution into another. Specifically, it learns to align trajectories between a prior noise distribution that we can easily sample from, to a data distribution (e.g., the distribution of real-world images or videos). We begin by presenting the definition of trajectories.

**Definition 2.1** (Trajectory)**.** *Let $t \in [0, 1]$ represent the timestep. Let $\alpha_t, \beta_t : [0, 1] \to \mathbb{R}$ denote interpolation functions, where $\alpha_0 = \beta_1 = 1$ and $\alpha_1 = \beta_0 = 0$. The prior distribution is $d$-dimensional multivariate Gaussian $\epsilon \sim \mathcal{N}(\mu, \sigma^2 I_d)$ and the data distribution is $x \sim p_{\text{data}}(x)$. Then, we define the trajectory $z_t$ as follows: $z_t := \alpha_t x + \beta_t \epsilon$.*

Considering both endpoints of the trajectory, we can simply obtain the following fact.

**Fact 2.2.** *Let $z_t$ be defined as in Definition 2.1. Then, we have $z_0 \sim p_{\text{data}}$ and $z_1 \sim \mathcal{N}(\mu, \sigma^2 I_d)$.*

Next, we define two velocity fields between the noise distribution and the data distribution. We first define the conditional velocity for one specific sample.

**Definition 2.3** (Conditional velocity, implicit in page 4 on (Lipman et al., 2023))**.** *Let $t \in [0, 1]$ denote the timestep. The trajectory $z_t \in \mathbb{R}^d$ is defined in Definition 2.1. Then, we define the conditional velocity $v_t$ as follows: $v_t := \frac{\mathrm{d}z_t}{\mathrm{d}t}$.*

Now, we define the marginal velocity, which is the expected velocity field for all possible samples.

**Definition 2.4** (Marginal velocity, implicit in page 3 on (Lipman et al., 2023)). *Let $t \in [0,1]$ denote the timestep. The trajectory $z_t \in \mathbb{R}^d$ is defined in Definition 2.1, and the conditional velocity $v_t \in \mathbb{R}^d$ defined in Definition 2.3. Then, we define the marginal velocity as follows: $v(z_t, t) := \mathbb{E}_{v_t \sim p_t(v_t|z_t)}[v_t]$.*

### 2.3 SECOND-ORDER FLOW MATCHING

Recently, a wide range of works (Liang et al., 2025; Chen et al., 2025a;b) have shown that flow matching models can gain advantages from parameterizing both velocity fields and acceleration fields with respect to the trajectory. Specifically, this benefit comes from the reduction of truncation error in numerical integration. For the transformation from trajectory $z_r$ to $z_t$, first-order methods rely on linear interpolation, which yields an error of $O((t-r)^2)$. In contrast, incorporating acceleration enables a second-order Taylor approximation that reduces the error to $O((t-r)^3)$, allowing for larger step sizes during inference. We provide a more detailed discussion on this motivation in Section B.

**Basic Concepts.** These models are built on the following formulations of acceleration fields. Similar to first-order flow matching, we first define the conditional acceleration.

**Definition 2.5** (Conditional acceleration). *Let $t \in [0,1]$ denote the timestep. The trajectory $z_t \in \mathbb{R}^d$ is defined in Definition 2.1. Then, we define the conditional acceleration $a_t$ as follows: $a_t := \frac{\mathrm{d}^2 z_t}{\mathrm{d}t^2}$.*

Computing the second-order derivative, we can rewrite the conditional acceleration as follows.

**Fact 2.6.** *For any $x \sim p_{\mathrm{data}}$ and $\epsilon \sim \mathcal{N}(\mu, \sigma^2 I_d)$, we have $a_t = \frac{\mathrm{d}^2 \alpha_t}{\mathrm{d}t^2} x + \frac{\mathrm{d}^2 \beta_t}{\mathrm{d}t^2} \epsilon$.*

Next, we define the marginal acceleration for the expected acceleration field for all possible samples.

**Definition 2.7** (Marginal acceleration). *Let $t \in [0,1]$ denote the timestep. The trajectory $z_t \in \mathbb{R}^d$ is defined in Definition 2.1, and the conditional acceleration $a_t \in \mathbb{R}^d$ defined in Definition 2.5. Then, we define the marginal velocity as follows: $a(z_t, t) := \mathbb{E}_{a_t \sim p_t(a_t|z_t)}[a_t]$.*

**Training.** To train these high-order flow-matching models, the loss function is computed for both velocity and acceleration fields.

**Definition 2.8** (Second-order flow matching loss). *Let $t \in [0,1]$ denote the timestep. The trajectory $z_t \in \mathbb{R}^d$, marginal velocity $v(z_t, t) \in \mathbb{R}^d$, and marginal acceleration $a(z_t, t) \in \mathbb{R}^d$ are defined in Definitions 2.1, 2.4, and 2.7, respectively. To learn the marginal velocity and acceleration fields, neural networks $u_{1,\theta_1}$ and $u_{2,\theta_2}$ are trained to minimize the following loss function: $L_{\mathrm{SO}}(\theta) := \mathbb{E}_{t,z_t}[\|u_{1,\theta_1}(z_t, t) - v(z_t, t)\|_2^2] + \mathbb{E}_{t,z_t}[\|u_{2,\theta_2}(z_t, t) - a_t(z_t, t)\|_2^2]$, where $t \sim \mathsf{Uniform}[0,1]$ and $z_t \sim p_t(z_t)$.*

Similar to first-order flow matching models, the original flow matching loss is still intractable. Thus, it is more practical to compute the following alternative loss function.

**Definition 2.9** (Conditional second-order flow matching loss). *Let $t \in [0,1]$ denote the timestep. The trajectory $z_t \in \mathbb{R}^d$, conditional velocity $v_t \in \mathbb{R}^d$, and conditional acceleration $a(z_t, t) \in \mathbb{R}^d$ are defined in Definitions 2.1, 2.3, and 2.5, respectively. To learn the conditional velocity and acceleration fields, neural networks $u_{1,\theta_1}$ and $u_{2,\theta_2}$ are trained to minimize the following loss function: $L_{\mathrm{CSO}}(\theta) := \mathbb{E}_{t,x,\epsilon}[\|u_{1,\theta_1}(z_t, t) - v_t\|_2^2] + \mathbb{E}_{t,x,\epsilon}[\|u_{2,\theta_2}(z_t, t) - a_t\|_2^2]$, where $t \sim \mathsf{Uniform}[0,1]$, $x \sim p_{\mathrm{data}}(x)$, and $\epsilon \sim \mathcal{N}(\mu, \sigma^2 I_d)$.*

## 3 FEASIBILITY OF SECOND-ORDER MeanFlow

In this section, we first provide a formal formulation of the MeanFlow model in Section 3.1, and then introduce our main results on the feasibility of high-order MeanFlow in Section 3.2.

### 3.1 MeanFlow

Recently, MeanFlow (Geng et al., 2025a) emerges as a promising variant of flow matching models, which exhibits both sampling efficiency and simple implementation.

**Average Velocity.** MeanFlow (Geng et al., 2025a) introduces a concept: the average velocity, which stands in contrast to the instantaneous velocity modelled in traditional Flow Matching. While Flow Matching's velocity describes motions at a specific moment, the average velocity captures the overall movement between two points in time.

**Definition 3.1** (Average velocity, implicit in page 3 of (Geng et al., 2025a)). *Let $t, r \in [0, 1]$ denote two different timesteps. The marginal velocity $v(z_\tau, \tau)$ is defined in Definition 2.4. Then, we define the average velocity between $t$ and $r$ as follows:*

$$\overline{v}(z_t, r, t) := \frac{1}{t - r} \int_r^t v(z_\tau, \tau) \mathrm{d}\tau. \tag{1}$$

**Consistency Condition.** The average velocity satisfies an important consistency condition, which ensures the efficient sampling of the training flow matching models (Song et al., 2023a; Song & Dhariwal, 2024; Geng et al., 2025a). We first introduce the definition of consistency functions.

**Definition 3.2** (Consistency function, implicit in page 3 of (Song et al., 2023a)). *Let $\delta \in (0, 1)$ be a small constant. For any timestep $t \in [0, 1]$ and trajectory $z_t \in \mathbb{R}^d$ defined in Definition 2.1, we say a vector-valued function $f : \mathbb{R}^d \times [0, 1] \to \mathbb{R}^d$ is a consistency function if it meets the conditions below:* **(i) Self-Consistency:** *For any $t_1, t_2 \in [\delta, 1]$, we have $f(z_{t_1}, t_1) = f(z_{t_2}, t_2)$.* **(ii) Boundary Condition:** $f(z_\delta, \delta) = z_\delta$.

The consistency condition in MeanFlow generalizes the definition of the consistency function, which results in the following definition.

**Definition 3.3** (Generalized consistency function, implicit in pages 3 of (Geng et al., 2025a)). *For two arbitrary timesteps $t, r \in [0, 1]$ ($r \leq t$) and any time-dependent vector trajectory $z_t \in \mathbb{R}^d$, we say a vector-valued function $f : \mathbb{R}^d \times [0, 1] \times [0, 1] \to \mathbb{R}^d$ is a generalized consistency function for $z_t$ if it meets the conditions below:* **(i) Self-Consistency:** *For any $s \in [r, t]$, we have $(t - r)f(z_t, r, t) = (s - r)f(z_s, r, s) + (t - s)f(z_t, s, t)$.* **(ii) Boundary Condition:** $f(z_r, r, r) = z_r$ and $f(z_t, t, t) = z_t$.

**Remark 3.4.** *The consistency condition of* MeanFlow *in Definition 3.3 generalizes the original consistency condition in Definition 3.2 by extending the trajectory $z_t$ to any time-dependent vector-valued function. The generalized consistency is defined on an arbitrary interval $[r, t]$ instead of fixed $[\delta, 1]$.*

Next, we show that the average velocity in MeanFlow satisfies the generalized consistency condition, which means that its sampling can be accelerated. We first show that the boundary condition is satisfied.

**Lemma 3.5** (Boundary condition of average velocity, informal version of Lemma D.1). *The relationship between the average velocity $\overline{v}$ (Definition 3.1) and the marginal velocity $v$ (Definition 2.4) is established by the boundary condition in Definition 3.3 as follows: $\overline{v}(z_r, r, r) = v(z_r, r)$ and $\overline{v}(z_t, t, t) = v(z_t, t)$.*

Next, we show that the self-consistency condition also holds for the average velocity.

**Lemma 3.6** (Consistency constraint of average velocity, informal version of Lemma D.3). *Let $r, t$ denote two different timesteps. The timestep $s$ is any intermediate time between $r$ and $t$. The marginal velocity $v$ is defined in Definition 2.4. The average velocity $\overline{v}$ defined in Definition 3.1 satisfies an additive consistency constraint: $(t - r)\overline{v}(z_t, r, t) = (s - r)\overline{v}(z_s, r, s) + (t - s)\overline{v}(z_t, s, t)$.*

Combining both lemmas above, we can conclude that the average velocity satisfies the generalized consistency condition.

**Theorem 3.7** (Consistency of average velocity). *The average velocity $\overline{v}(z_t, r, t)$ defined in Definition 3.1 is a generalized consistency function of $v(z_t, t)$ in Definition 2.4.*

*Proof.* This directly follows from Lemma 3.5 and Lemma 3.6. $\square$

**Training.** To train the MeanFlow models, we apply a similar loss function as the vanilla flow-batching models.

**Definition 3.8** (Ideal first-order MeanFlow loss). $L^*_{\mathrm{FOM}}(\theta) := \mathbb{E}_{t, z_t}[\|u_{1, \theta_1}(z_t, r, t) - \overline{v}(z_t, r, t)\|_2^2]$, *where $t \sim \mathsf{Uniform}[0, 1]$, $r \sim \mathsf{Uniform}[0, 1]$ and $z_t \sim p_t(z_t)$. Here FOM denotes first-order MeanFlow.*

**Remark 3.9.** *The* MeanFlow *model aims to approximate the average velocity* $\overline{v}(z_t, r, t)$ *defined in Definition 3.1 with a neural network* $u_\theta(z_t, r, t)$. *This strategy offers a distinct advantage: provided an accurate approximation, we can reconstruct the entire flow trajectory with evaluation of* $u_\theta(\epsilon, 0, 1)$. *Empirically, this makes the approach significantly more suitable for single or few-step generation, as it eliminates the need for explicit time integral approximation during inference, which is a challenge inherent in models of instantaneous velocity.*

**Remark 3.10.** *Nevertheless, directly using the average velocity as an objective function for training is impractical because it necessitates integral evaluation during the training phase. The core innovation of meanflow lies in transforming the definitional equation of the average velocity to an optimization target that can be effectively trained, even with access to only instantaneous velocity.*

Next, we present a practical loss function for MeanFlow, which carefully tackles the computation of mean velocities and conditional expectations.

**Definition 3.11** (First-order MeanFlow loss, implicit in page 5 on (Geng et al., 2025a))**.** *Let* $\text{sg}(\cdot)$ *denote the stopping gradient operation. Let* $t, r \in [0, 1]$ *denote the timestep.* $z_t$ *is defined in Definition 2.1. A neural network* $u_{1,\theta_1}$ *is trained to minimize the following loss function:* $L_{\text{FOM}}(\theta) := \mathbb{E}_{r,t,x,\epsilon}[\|u_{1,\theta_1}(z_t, r, t) - \text{sg}(\overline{v}(z_t, r, t))\|_2^2]$, *where* $t \sim \text{Uniform}[0, 1], r \sim \text{Uniform}[0, 1], x \sim p_{\text{data}}(x), \epsilon \sim \mathcal{N}(\mu, \sigma^2 I_d)$, *and the mean velocity can be computed as:*

$$\overline{v}(z_t, r, t) = v(z_t, t) - (t - r)\frac{\mathrm{d}}{\mathrm{d}t}\overline{v}(z_t, r, t). \tag{2}$$

**Remark 3.12.** *To facilitate optimization and prevent "double backpropagation" through the Jacobian-vector product, a stop-gradient (sg) operation is applied to* $\overline{v}(z_t, r, t)$ *in the loss function, consistent with previous works (Song et al., 2023a; Song & Dhariwal, 2024; Geng et al., 2025b; Lu & Song, 2025; Frans et al., 2025).*

The loss function in Definition 3.11 has two desirable properties, which ensure its effectiveness and computational efficiency. First, we show the effectiveness result.

**Theorem 3.13** (Equivalence of MeanFlow loss functions, informal version of Theorem D.5)**.** *The first-order* MeanFlow *loss function* $L_{\text{FOM}}(\theta)$ *in Definition 3.11 is equivalent to the ideal first-order* MeanFlow *loss function* $L_{\text{FOM}}^*(\theta)$ *in Definition 3.8.*

Next, we show that the first-order MeanFlow loss can be computed efficiently with JVPs.

**Theorem 3.14** (Efficiency of MeanFlow loss, informal version of Theorem D.6)**.** *The loss function* $L_{\text{FOM}}(\theta)$ *in Definition 3.11 can be evaluated efficiently with the conditional velocity* $v_t$ *in Definition 2.3 and Jacobian-Vector Products (JVPs) of the neural network* $u_{1,\theta_1}$, *i.e.,* $\overline{v}(z_t, r, t) = v_t - (t - r)(v_t \partial_z u_\theta + \partial_t u_\theta)$.

**Remark 3.15.** *In contemporary programming libraries, the JVP computations in Theorem 3.14 can be performed efficiently using JVP interfaces, such as* `torch.func.jvp` *in PyTorch or* `jax.jvp` *in JAX, which ensures convenient implementation and fast computation.*

## 3.2 MAIN RESULTS ON SECOND-ORDER MeanFlow

Building on the concept of average velocity, we introduce average acceleration $\overline{a}$, which extends the vanilla MeanFlow to a high-order case.

**Definition 3.16.** *Let* $t, r$ *denote two different timesteps. For any marginal acceleration* $a(z_t, t)$ *defined in Definition 2.7, we define the average acceleration as follows:*

$$\overline{a}(z_t, r, t) := \frac{1}{t - r} \int_r^t a(z_\tau, \tau) \mathrm{d}\tau \tag{3}$$

**Consistency Condition.**  We first show our key result on the consistency conditions. We begin by proving that the boundary condition is satisfied for the average acceleration.

**Lemma 3.17** (Boundary conditions of average acceleration, informal version of Lemma D.2)**.** *Given the average acceleration* $\overline{a}$ *defined in Definition 3.16 and the marginal acceleration* $a$ *defined in Definition 2.7,* $\overline{a}$ *converges to* $a$ *as the time interval shrinks to zero. Specifically:* $\overline{a}(z_r, r, r) = a(z_r, r)$ *and* $\overline{a}(z_t, t, t) = a(z_t, t)$.

Similar to the first-order MeanFlow results, the consistency constraint also holds for the average acceleration.

**Lemma 3.18** (Consistency constraint of average acceleration, informal version of Lemma D.4). *Let $r, s,$ and $t$ be three distinct time steps, such that $s$ lies between $r$ and $t$. Given the marginal acceleration $a$ (as defined in Definition 2.7), the average acceleration $\overline{a}$ (as defined in Definition 3.16) satisfies the following additive consistency constraint: $(t - r)\overline{a}(z_t, r, t) = (s - r)\overline{a}(z_s, r, s) + (t - s)\overline{a}(z_t, s, t)$.*

Combining both lemmas above, we obtain the consistency result for the second-order MeanFlow.

**Theorem 3.19** (Consistency of second-order MeanFlow). *The average acceleration $\overline{a}(z_t, r, t)$ defined in Definition 3.16 is a generalized consistency function of $a(z_t, t)$ in Definition 2.7.*

*Proof.* This directly follows from Lemma 3.17 and Lemma 3.18. $\qquad\square$

**Training.** Similar to the original MeanFlow, we show that the second-order extension also enjoys effective and efficient training. First, we show the ideal form of the second-order MeanFlow loss, which matches the original flow matching loss function but is intractable in practice.

**Definition 3.20** (Ideal second-order MeanFlow loss). *This ideal loss simultaneously trains two networks: $u_{1,\theta_1}$ to predict the mean velocity $\overline{v}(z_t, r, t)$ and $u_{2,\theta_2}$ to predict the mean acceleration $\overline{a}(z_t, r, t)$. The loss is the sum of their respective mean squared errors, averaged over the distributions of $t, r,$ and $z_t$. $L^*_{\mathrm{SOM}}(\theta) := \mathbb{E}_{t,r,z_t}[\|u_{1,\theta_1}(z_t, r, t) - \overline{v}(z_t, r, t)\|_2^2] + \mathbb{E}_{t,r,z_t}[\|u_{2,\theta_2}(z_t, r, t) - \overline{a}(z_t, r, t)\|_2^2]$, where $t \sim \mathsf{Uniform}[0, 1], r \sim \mathsf{Uniform}[0, 1],$ and $z_t \sim p_t(z_t)$.*

The intractability of the ideal second-order MeanFlow loss has necessitated the following loss function:

**Definition 3.21** (Second-order MeanFlow loss). *Let $\mathrm{sg}(\cdot)$ denote the stopping gradient operation. Let $t, r \in [0, 1]$ denote the timestep. $z_t$ is defined in Definition 2.1. Two neural networks $u_{1,\theta_1}$ and $u_{2,\theta_2}$ are trained to minimize the following loss function:*

$$L_{\mathrm{SOM}}(\theta) := \mathop{\mathbb{E}}_{t,r,x,\epsilon}[\|u_{1,\theta_1}(z_t, r, t) - \overline{v}(z_t, r, t)\|_2^2] + \mathop{\mathbb{E}}_{t,r,x,\epsilon}[\|u_{2,\theta_2}(z_t, r, t) - \overline{a}(z_t, r, t)\|_2^2],$$

*where $t \sim \mathsf{Uniform}[0, 1], r \sim \mathsf{Uniform}[0, 1],$ $x \sim p_{\mathrm{data}}(x), \epsilon \sim \mathcal{N}(\mu, \sigma^2 I_d),$ and the mean velocity and acceleration can be computed as: $\overline{v}(z_t, r, t) = v(z_t, t) - (t - r)\frac{\mathrm{d}}{\mathrm{d}t}\overline{v}(z_t, r, t), \overline{a}(z_t, r, t) = a(z_t, t) - (t - r)\frac{\mathrm{d}}{\mathrm{d}t}\overline{a}(z_t, r, t).$*

The loss function above has two desirable properties. First, we show that the tractable second-order MeanFlow loss function is equivalent to the ideal loss function.

**Theorem 3.22** (Effectivenss of second-order MeanFlow, informal version of Theorem D.7). *The loss function $L_{\mathrm{SOM}}(\theta)$ in Definition 3.21 is equivalent to the ideal MeanFlow loss function $L^*_{\mathrm{SOM}}(\theta)$ in Definition 3.20.*

Next, we show that the second-order MeanFlow loss can be computed efficiently with JVPs.

**Theorem 3.23** (Efficiency of second-order MeanFlow, informal version of Theorem D.8). *The loss function $L_{\mathrm{SOM}}(\theta)$ in Definition 3.21 can be evaluated efficiently with the conditional velocity $v_t$ and conditional acceleration $a_t$ in and Jacobian-Vector Products (JVPs) of the neural networks $u_{1,\theta_1}$ and $u_{2,\theta_2}$, i.e., $\overline{v}(z_t, r, t) = v_t - (t - r)(v_t \partial_z u_{1,\theta_1} + \partial_t u_{1,\theta_1}), \overline{a}(z_t, r, t) = a_t - (t - r)(a_t \partial_z u_{2,\theta_2} + \partial_t u_{2,\theta_2}).$*

# 4 CIRCUIT COMPLEXITY OF SECOND-ORDER MeanFlow

Circuit complexity serves as a fundamental theoretical tool for analyzing the expressiveness of machine learning models. Recent works have established that Transformers without chain-of-thought prompting (Wei et al., 2022) belong to the $\mathsf{TC}^0$ class (Li et al., 2024), implying they are unable to solve $\mathsf{NC}^1$-hard problems unless the open conjecture $\mathsf{TC}^0 = \mathsf{NC}^1$ holds (Vollmer, 1999). In this section, we extend this theoretical perspective to the context of generative modeling. We prove that the sampling processes of MeanFlow models, with or without high-order flow augmentations, strictly belong to the $\mathsf{TC}^0$ class under mild assumptions.

### 4.1 MeanFlow FORMATION

In this section, we provide the formal definitions for MeanFlow's sampling.

**Definition 4.1** (MeanFlow $T$-step sampling). *Given $T \in \mathbb{Z}_+$ as the total iteration of sampling, and arbitrary $\{t_i\}_{i=1}^T$ that satisfy $t_i > t_{i+1}$ for all $i \in \{1, \ldots, T-1\}$ as timestep scheduler. Let $Z_1 \in \mathbb{F}_p^{n \times d}$ become the initial feature map for MeanFlow.* Let ViT *denote Vision Transformer. Then according to Fact C.10, the sampling process of MeanFlow is given by*

$$\mathsf{Z}_{t_i} := \mathsf{Z}_{t_{i-1}} + (t_i - t_{i-1})\mathsf{ViT}(\mathsf{Z}_{t_{i-1}} \,\|\, t_i \mathbf{1}_n \,\|\, t_{i-1} \mathbf{1}_n)_{*,1:d} \in \mathbb{R}^n,$$

*where $\|$ denotes the concatenation operation. We use $\mathsf{MF}(Z_{t_1}, t_1, \cdots, t_T) := Z_{t_T}$ to denote a $T$-step sampling of MeanFlow as a function.*

### 4.2 SECOND-ORDER MeanFlow FORMATION

In this section, we provide the formal definitions of Second-order MeanFlow's sampling.

**Definition 4.2** (Second-order MeanFlow $k$-step sampling). *Given $T \in \mathbb{Z}_+$ as the total iteration of sampling, and arbitrary $\{t_i\}_{i=1}^T$ that satisfy $t_i > t_{i+1}$ for all $i \in \{1, \ldots, T-1\}$ as timestep scheduler. Let $Z_0 \in \mathbb{F}_p^{n \times d}$ become the initial feature map for MeanFlow. Then according to Fact C.14, the sampling process of Second-order MeanFlow is given by*

$$\mathsf{Z}_{t_i} := \mathsf{Z}_{t_{i-1}} + (t_i - t_{i-1})\mathsf{ViT}_1(\mathsf{Z}_{t_{i-1}} \,\|\, t_i \mathbf{1}_n \,\|\, t_{i-1} \mathbf{1}_n)_{*,1:d}$$
$$+ \frac{1}{2}(t_i - t_{i-1})^2 \mathsf{ViT}_2(\mathsf{Z}_{t_{i-1}} \,\|\, t_i \mathbf{1}_n \,\|\, t_{i-1} \mathbf{1}_n)_{*,1:d} \in \mathbb{R}^{n \times d},$$

*where $\mathsf{ViT}_1$ and $\mathsf{ViT}_2$ are two ViTs, and $\|$ denotes the concatenation operation. We use $\mathsf{SMF}(Z_{t_1}, t_1, \cdots, t_T) := Z_{t_T}$ to denote a $T$-step sampling of Second-order MeanFlow.*

### 4.3 CIRCUIT COMPLEXITY OF MeanFlow WITH EULER SOLVER

In this section, we show the circuit complexity of sampling vanilla MeanFlow and its second-order version with Euler solver both belong to $\mathsf{TC}^0$.

We first prove the original MeanFlow sampling with $T$-step Euler solver belongs to $\mathsf{TC}^0$.

**Theorem 4.3** (MeanFlow sampling with $T$-step Euler solver belongs to $\mathsf{TC}^0$, informal version of Theorem E.8). *Given $T \in \mathbb{Z}_+$ as the total iteration of sampling, and arbitrary $\{t_i\}_{i=1}^T$ that satisfy $t_i > t_{i+1}$ for all $i \in \{1, \ldots, T-1\}$ as timestep scheduler. Assume the precision $p \leq \mathrm{poly}(n)$, the number of transformer layers $m = O(1)$, and $T = O(1)$. Then we can use a size bounded by $\mathrm{poly}(n)$ and constant depth $T(m(17d_{\mathrm{std}} + 8d_\oplus + 2d_{\mathrm{sqrt}} + d_{\exp}) + 5d_{\mathrm{std}})$ uniform threshold circuit to simulate the MeanFlow $T$-step sampling defined in Definition 4.1.*

We then prove the sampling circuit complexity of Second-order MeanFlow with $T$-step Euler solver belongs to $\mathsf{TC}^0$.

**Theorem 4.4** (Second-order MeanFlow sampling with $T$-step Euler solver belongs to $\mathsf{TC}^0$, informal version of Theorem E.9). *Given $T \in \mathbb{Z}_+$ as the total iteration of sampling, and arbitrary $\{t_i\}_{i=1}^T$ that satisfy $t_i > t_{i+1}$ for all $i \in \{1, \ldots, T-1\}$ as timestep scheduler. Assume the precision $p \leq \mathrm{poly}(n)$, the number of transformer layers $m = O(1)$, and $T = O(1)$. Then we can use a size bounded by $\mathrm{poly}(n)$ and constant depth $2T(m(17d_{\mathrm{std}} + 8d_\oplus + 2d_{\mathrm{sqrt}} + d_{\exp}) + 5d_{\mathrm{std}}) + Td_{\mathrm{std}}$ uniform threshold circuit to simulate the MeanFlow $T$-step sampling defined in Definition 4.2.*

## 5 PROVABLY EFFICIENT CRITERIA

Recent research has established provable efficiency results for LLMs, identifying conditions under which attention mechanisms can be accelerated (Alman & Song, 2023; 2024a;b; 2025a;b). We extend this theoretical framework to the generative modeling setting in vision. We demonstrate that our high-order MeanFlow models can be computed efficiently, with desirable inference speed guaranteed in theory.

In this section, we first provide an approximate attention computation in Section 5.1. Then, we present a running time analysis for second-order MeanFlow in Section 5.2. We also present the running time analysis for its fast version in Section 5.3. Next, we analyze the error propagation of the fast second-order MeanFlow in Section 5.4. Finally, we show the existence result of a fast algorithm that computes the second-order MeanFlow architecture in Section 5.5.

## 5.1 APPROXIMATE ATTENTION COMPUTATION

In this section, we introduce approximate attention computation, which accelerates the attention layer's computation.

**Definition 5.1** (Approximate attention computation, Definition 1.2 in (Alman & Song, 2023)). *Let $n, d > 0$ be positive constants representing the number of tokens and the dimension of the embeddings, respectively. We are given three matrices: the Query matrix $Q \in \mathbb{R}^{n \times d}$, the Key matrix $K \in \mathbb{R}^{n \times d}$, and the Value matrix $V \in \mathbb{R}^{n \times d}$. These matrices are guaranteed to have a bounded infinity norm: $\|Q\|_\infty \leq R, \|K\|_\infty \leq R, \|V\|_\infty \leq R$ for some known constant $R > 0$.*

*For any attention layer $\mathsf{Attn}(Q, K, V)$, we use an approximate attention layer, denoted by $\mathsf{AAttC}(n, d, R, \delta)$, which produces an output matrix $O \in \mathbb{R}^{n \times d}$ that approximates the true output with a guaranteed error bound $\delta = 1/\mathrm{poly}(n)$: $\|O - \mathsf{Attn}(Q, K, V)\|_\infty \leq 1/\mathrm{poly}(n)$.*

The next lemma specifies the time complexity for the AAttC method.

**Theorem 5.2** (Time complexity of approximate attention computation, Theorem 1.4 of (Alman & Song, 2023)). *We define $\mathsf{AAttC}$ in Definition 5.1. Let the parameters for $\mathsf{AAttC}$ be set as follows: an embedding dimension of $d = O(\log n)$, $R = \Theta(\sqrt{\log n})$, and an approximation tolerance of $\delta = 1/\mathrm{poly}(n)$. Based on these conditions, the time complexity is: $\mathcal{T}(n, n^{o(1)}, d) = n^{1+o(1)}$.*

## 5.2 INFERENCE RUNTIME OF SECOND-ORDER MeanFlow

This section analyzes the running time of the second-order MeanFlow inference pipeline.

**Lemma 5.3** (Sampling runtime of Second-order MeanFlow, informal version of Lemma F.1). *Consider the original second-order MeanFlow inference pipeline. The input is a tensor $\mathsf{X} \in \mathbb{R}^{h \times w \times c}$, where the height $h$ and width $w$ are both equal to $n$, and the number of channels $c$ is on the order of $O(\log n)$. The interpolated state at time $t \in [0, 1]$ is denoted by $\mathsf{F}^t$, with $\mathsf{F}^1$ representing the final state. The model architecture consists of attention (Attn), MLP (MLP($\cdot, c, d$)), and Layer Normalization (LN) layers. Based on these conditions, the inference time complexity of second-order MeanFlow is bounded by $O(n^{4+o(1)})$.*

## 5.3 INFERENCE RUNTIME OF FAST SECOND-ORDER MeanFlow

This section applies the conclusions of (Alman & Song, 2023) to the second-order MeanFlow architecture. Specifically, we replace each of its attention modules with the approximate attention defined in Definition 5.1.

**Lemma 5.4** (Sampling runtime of fast Second-order MeanFlow, informal version of Lemma F.2). *Consider the fast second-order MeanFlow inference pipeline. It takes an input tensor $\mathsf{Z}_1 \in \mathbb{R}^{h \times w \times c}$, where the height $h = n$, width $w = n$, and the number of channels $c = O(\log n)$. The model architecture consists of attention (AAttC), MLP (MLP($\cdot, c, d$)), and Layer Normalization (LN) layers. The interpolated state at time $t \in [0, 1]$ is denoted by $\mathsf{Z}_t$, with $\mathsf{Z}_0$ as the final state. Based on these conditions, the inference time complexity of fast second-order MeanFlow is bounded by $O(n^{2+o(1)})$.*

## 5.4 ANALYSIS OF ERROR PROPAGATION

This section shows an error analysis for the approximate attention applied to the second-order MeanFlow model. We conduct the error analysis between approximate attention computation in Definition 5.1 and vanilla attention. We begin by analyzing the errors for the attention matrices.

**Lemma 5.5** (Error analysis of $\mathsf{AAttC}(\mathsf{Z}'_1)$ and $\mathsf{Attn}(\mathsf{Z}_1)$, Lemma B.4 in (Ke et al., 2025)). *Let $\mathsf{AAttC}(\mathsf{Z}'_1)$ be an input tensor and $\mathsf{Attn}(\mathsf{Z}_1)$ be its approximation, satisfying an element-wise error bound $\|\mathsf{Z}'_1 - \mathsf{Z}_1\| \leq \epsilon$ for some $\epsilon \in (0, 0.1)$. Let $\mathsf{Attn}(\cdot)$ denote the standard attention layer defined*

*in Definition C.28 and* $\mathsf{AAttC}(\cdot)$ *represent the approximate attention layer defined in Definition 5.1, which utilizes a polynomial $f$ of degree $g$ constructed from low-rank matrices $U, V \in \mathbb{R}^{hw \times k}$. Assume the entries of the query, key, and value matrices are bounded in magnitude by a constant $R > 1$. Then, the element-wise error between the approximated attention output on the approximated input and the standard attention output on the original input is bounded as follows:* $\|\mathsf{AAttC}(\mathsf{Z}'_1) - \mathsf{Attn}(\mathsf{Z}_1)\|_\infty \leq O(kR^{g+1}c) \cdot \epsilon$, *where we extend the $\ell_\infty$ norm to apply tensors.*

Next, we analyze the MLP layer, which is part of the MeanFlow layers.

**Lemma 5.6** (Error analysis of MLP layer, Lemma C.3 of (Gong et al., 2025a)). *Let* $\mathsf{MLP}(\cdot, c, d)$ *be the MLP layer as defined in Definition C.30. Let* $\mathsf{AAttC}(\mathsf{Z}'_1)$ *be an input tensor and* $\mathsf{Attn}(\mathsf{Z}_1)$ *be its approximation, satisfying an element-wise error bound* $\|\mathsf{Z}'_1 - \mathsf{Z}_1\| \leq \epsilon$ *for some* $\epsilon \in (0, 0.1)$. *Assume the entries of the MLP's internal weight matrices are bounded in magnitude by a constant $R > 1$. Then, the element-wise error between the MLP outputs for the approximated and original inputs is bounded as follows:* $\|\mathsf{MLP}(\mathsf{Z}'_1) - \mathsf{MLP}(\mathsf{Z}_1)\|_\infty \leq cR\epsilon$, *where we abuse the $\ell_\infty$ norm in its tensor form for clarity.*

Then, we compute the error bound for the MeanFlow layer under fast attention computation.

**Lemma 5.7** (Error bound between fast second-order MeanFlow layer and second-order MeanFlow layer, informal version of Lemma F.3). *For the error analysis of the second-order* MeanFlow *Layer, we establish the following conditions. Let* $\mathsf{Z}_1 \in \mathbb{R}^{h \times w \times c}$ *be the input tensor and let* $\mathsf{Z}'_1 \in \mathbb{R}^{h \times w \times c}$ *be its approximation, such that the approximation error is bounded by* $\|\mathsf{Z}_1 - \mathsf{Z}'_1\|_\infty \leq \epsilon$ *for some small constant $\epsilon > 0$.*

*The analysis considers interpolated inputs* $\mathsf{Z}_t, \mathsf{Z}_{\mathrm{fast},t} \in \mathbb{R}^{h \times w \times c}$ *over a time step $t \in [0, 1]$. These inputs are processed by a standard second-order* MeanFlow *layer,* $\mathsf{SMF}(\cdot, \cdot, \cdot)$, *and a fast variant,* $\mathsf{SMF}_{\mathrm{fast}}(\cdot, \cdot, \cdot)$, *where the fast variant substitute* $\mathsf{Attn}$ *operation with* $\mathsf{AAttC}$ *(Definition 5.1). The approximation is achieved via a polynomial $f$ of degree $g$ and low-rank matrices $U, V \in \mathbb{R}^{hw \times k}$.*

*We assume all matrix entries are bounded by a constant $R > 1$. Crucially, we also make assumption that the LayerNorm function,* $\mathsf{LN}(\cdot)$ *defined in Definition C.31, does not exacerbate error propagation; that is, if* $\|\mathsf{Z}'_1 - \mathsf{Z}_1\|_\infty \leq \epsilon$, *then it follows that* $\|\mathsf{LN}(\mathsf{Z}'_1) - \mathsf{LN}(\mathsf{Z}_1)\|_\infty \leq \epsilon$. *Then, we have* $\|\mathsf{SMF}_{\mathrm{fast}}(\mathsf{Z}_{\mathrm{fast},t}, t, r) - \mathsf{SMF}(\mathsf{Z}_t, t, r)\|_\infty \leq O(c^2 k R^{g+2})\epsilon$.

## 5.5 ALMOST QUADRATIC TIME ALGORITHM

This section shows a theorem on the existence of a almost quadratic-time algorithm that approximates the second-order MeanFlow architecture with guaranteed a additive error bound.

**Theorem 5.8** (Almost quadratic time algorithm). *Suppose $d = O(\log n)$ and $R = o(\sqrt{\log n})$. Then, there exists an algorithm that can approximate the second-order* MeanFlow *architecture with an additive error of at most $1/\mathrm{poly}(n)$ in $O(n^{2+o(1)})$ time.*

*Proof.* The result follows directly by combining Lemma 5.4 and Lemma 5.7. □

## 6 CONCLUSION

We have presented second-order MeanFlow as a principled generalization of MeanFlow that models not only average velocities but also average accelerations, thereby enriching the dynamic expressivity of simulation-free generative flows. Our theoretical analysis demonstrates that (i) average acceleration fields obey a consistency condition enabling single-step inference, (ii) the entire sampling mechanism resides within the $\mathsf{TC}^0$ circuit complexity class, and (iii) fast approximate attention techniques ensure provable efficiency with only $1/\mathrm{poly}(n)$ error in $O(n^{2+o(1)})$ time. These findings confirm that high-order flow matching is both theoretically sound and computationally tractable. In future work, we plan to empirically validate second-order MeanFlow on large-scale image and video benchmarks, explore extensions such as discrete flow matching and adaptive timestepping. Our results open a promising direction for the development of richer, faster generative models grounded in high-order differential dynamics.

ETHICS STATEMENT

This paper investigates the theoretical feasibility of applying high-order augmentations to MeanFlow generative models. It highlights a promising direction for future research while addressing fundamental limitations related to circuit complexity and criteria for provable efficiency. As this study is purely theoretical, we do not foresee any negative societal implications.

REPRODUCIBILITY STATEMENT

Since this paper is entirely theoretical, we do not foresee any reproducibility issues related to experimental results. For theoretical reproducibility, all assumptions are explicitly stated within theorems and lemmas. Proofs are provided either directly following the statements or in the corresponding sections of the appendix. For any statement without a proof in the main text, we include a clear reference to its location in the appendix.

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

# Appendix

**Roadmap.** In Section A, we introduce work that related to our research. In Section B, we discuss the motivation of the proposed high-order MeanFlow models. In Section C, we present some backgrounds of this paper. In Section D, we supplement the missing proofs in Section 3. In Section E, we provide the missing proofs in Section 4. In Section F, we show the missing proofs in Section 5.

## A    RELATED WORK

**Flow Matching.** Flow Matching architecture (Lipman et al., 2023; Albergo & Vanden-Eijnden, 2023; Liu et al., 2023b) has emerged as a powerful alternative to simulation-based Continuous Normalizing Flows (CNFs) (Chen et al., 2018), providing a simulation-free objective by learning to regress velocity fields along pre-specified probability paths from simple noise distributions to complex data distributions. Many works have already validated its capability on complicated real-world applications, e.g. image generation (Lipman et al., 2023; Esser et al., 2024; Frans et al., 2025; Huang et al., 2024) and video generation (Polyak et al., 2024; Jin et al., 2025; Cao et al., 2025b). (Tong et al., 2024) introduces Conditional Flow Matching (CFM) and its variant Optimal Transport Conditional Flow Matching (OT-CFM). CFM defines a class of simulation-free training objectives for CNFs, enabling efficient conditional generative modeling while accelerating both training and inference. OT-CFM further enhances this by approximating dynamic optimal transport without simulation, resulting in more stable and efficient learning. Building on these foundations, recent work has proposed several innovative extensions to flow matching. For instance, (Klein et al., 2023) introduces an equivariant extension of flow matching for physics-based generative tasks. (Haviv et al., 2025) presents Wasserstein Flow Matching, which generalizes traditional flow matching to different families of distributions, expanding its utility in areas such as computer graphics and genomics. (Gat et al., 2024) extends flow matching to discrete cases, which enables flow matching for discrete modeling, such as natural language. (Cao et al., 2025a) explores the integration of special relativity constraints into the flow matching framework. More recent advances on flow matching include flow matching on different geometry structures (Cheng et al., 2024; Chen & Lipman, 2024; Song et al., 2023b; Kapusniak et al., 2024), PDEs (Baldan et al., 2025; Fotiadis et al., 2024; Li et al., 2025b; Cheng et al., 2025), guided flow matching (Zheng et al., 2023; Haber et al., 2025; Cao et al., 2025b), benchmarking of generative models (Chen et al., 2026; Guo et al., 2025a;b;c), and computational limits (Long et al., 2025; Chen et al., 2025e).

**Circuit Complexity.** As a cornerstone of theoretical computer science, circuit complexity investigates the computational capabilities and limitations of Boolean circuit families. This framework has recently found increasing application in the analysis of machine learning models, providing a principled approach to characterize their inherent computational capabilities. Several circuit complexity classes are particularly relevant in this domain. Crucial to this study is the class $\mathsf{AC}^0$, which consists of decision problems solvable by constant-depth Boolean circuits with unbounded fan-in and the logic gates AND, OR, and NOT, capturing problems that are highly parallelizable using standard logic gates. $\mathsf{TC}^0$ extends $\mathsf{AC}^0$ by incorporating MAJORITY gates (also known as threshold gates). $\mathsf{NC}^1$ represents languages recognizable by circuits of $O(\log n)$ depth with bounded gate arity (Merrill et al., 2022). A well-established inclusion chain in this context is $\mathsf{AC}^0 \subset \mathsf{TC}^0 \subseteq \mathsf{NC}^1$, but whether $\mathsf{TC}^0 = \mathsf{NC}^1$ holds an open question (Vollmer, 1999; Arora & Barak, 2009). Recent breakthroughs in circuit complexity have offered a rigorous framework for evaluating the computational limits of various neural network architectures. In particular, attention has turned to Transformer models and two of their key derivatives: SoftMax-Attention Transformers (SMATs) and Average-Head Attention Transformers (AHATs). Prior work has shown that SMATs have been demonstrated to admit efficient simulations via $L$-uniform circuits with comparable depth and gate constraints (Liu et al., 2023a). Similarly, AHATs can be efficiently simulated by non-uniform threshold circuits of constant depth, placing them within the $\mathsf{TC}^0$ class (Merrill et al., 2022). These results have been further strengthened by subsequent studies establishing that both AHAT and SMAT architectures can be approximated using DLOGTIME-uniform $\mathsf{TC}^0$ circuits (Merrill & Sabharwal, 2023), thereby reinforcing the connection between Transformer models and low-depth threshold circuit classes. Circuit complexity has been proven to be an effective tool for showing the fundamental limitations of deep architectures. Its success has recently been extended to RoPE-Transformers (Chen et al., 2024b;

Yang et al., 2025; Chen et al., 2025c), State Space Models (Merrill et al., 2024; Chen et al., 2025d), and Graph Neural Networks (GNNs) (Barceló et al., 2020; Li et al., 2025a).

## B  DISCUSSION

In this section, we discuss the motivation of the proposed high-order MeanFlow models, and show the practical implications of our theoretical results.

**Why High-order** MeanFlow **Matters?** Modern generative models rely on solving differential equations that describe how data transforms over time. In practice, these flows are typically solved using first-order methods like Euler's method, which capture only the immediate direction of change while ignoring higher-order structure.

Consider any smooth trajectory $z(t)$. Taylor's theorem tells us we can expand around time $r$:

$$z_t = z_r + (t - r)z'_r + \frac{(t - r)^2}{2}z''_r + \frac{(t - r)^3}{6}z'''(\xi),$$

where $\xi \in (r, t)$.

First-order methods use only the linear terms: $z_t^{(1)} := z_r + (t - r)z'_r$, while second-order methods include the quadratic term: $z_t^{(2)} := z_r + (t - r)z'_r + \frac{(t-r)^2}{2}z''r$.

Then, the approximation errors are:

- First-order error:

$$
\begin{aligned}
E_1 &= z_t - z_t^{(1)} \\
&= \frac{(t - r)^2}{2}z''_r + \frac{(t - r)^3}{6}z'''(\xi) = O((t - r)^2).
\end{aligned}
$$

- Second-order error:

$$E_2 = z_t - z_t^{(2)} = \frac{(t - r)^3}{6}z'''(\xi) = O((t - r)^3).$$

This implies that second-order methods can achieve the same accuracy with larger steps, or better accuracy with the same computational cost. For generative models, this translates to fewer integration steps for sampling, improved numerical stability during training, and richer expressiveness through curvature information.

High-order MeanFlow leverages this insight to improve flow-based generative modeling by incorporating second-order derivatives into the flow approximation process.

**Practical Implications of Expressivity (Theorem 4.4).** Circuit complexity is a fundamental theoretical tool for analyzing the expressiveness of ML models. For example, Transformers without chain-of-thought prompting (Wei et al., 2022) belong to the $\mathsf{TC}^0$ class (Li et al., 2024), which makes it unable to solve $\mathsf{NC}^1$ hard unless an open conjecture $\mathsf{TC}^0 = \mathsf{NC}^1$ holds (Vollmer, 1999). In this work, we make a significant extension of this theory to the context of generative modeling, and find that MeanFlow models with or without high-order flow augmentations belong to the $\mathsf{TC}^0$ class in Theorem 4.4.

Thus, the key takeaway for practitioners is that, despite the strong modeling capability of high-order mean-flow models in capturing the distribution of image or video data, such architectures may not resolve the inherent limitations of their backbone models, such as ViT (Dosovitskiy et al., 2021) or DiT (Peebles & Xie, 2023). We suggest that practitioners may consider expressivity enhancement techniques in such models, such as looped Transformers (Giannou et al., 2023; Yang et al., 2024; De Luca & Fountoulakis, 2024; Saunshi et al., 2025), or improved positional encoding techniques (Yang et al., 2025).

**Practical Implications of Provable Efficient Criteria (Theorem 5.8).** Our key results in Theorem 5.8 show that the existence of an algorithm that is quadratic in the image size $n$ to compute the forward process of High-order MeanFlow models, and what condition is needed for the existence

of such efficient approximations. First, this suggests that our high-order MeanFlow models can be computed efficiently in practice, with desirable inference speed guaranteed in theory. Second, the condition $R = o(\sqrt{\log n})$ highlights that proper normalization may be needed to train this new type of model, since extremely large model weights may adversely affect the approximation error and make such efficient algorithms impossible. Our result extends previous provable efficiency results on LLMs (Alman & Song, 2023; 2024a;b; 2025a;b) and offers new insights within the generative modeling setting in vision.

## C  PRELIMINARY

This section provides the foundational background for the paper. In Section C.1, we introduce the principles of flow matching. In Section C.2, we present the basics of circuit complexity. In Section C.3, we show the definition of the Vision Transformer formulation.

### C.1  FLOW MATCHING

First, we compute the derivative of $z_t$ with respect to $t$, which yields the following fact.

**Fact C.1.** *For any $x \sim p_{\mathrm{data}}$ and $\epsilon \sim \mathcal{N}(\mu, \sigma^2 I_d)$, we have $v_t = \frac{\mathrm{d}\alpha_t}{\mathrm{d}t} x + \frac{\mathrm{d}\beta_t}{\mathrm{d}t} \epsilon$.*

Then, we provide two basic facts on the probability along the trajectory.

**Fact C.2** (Conditional probability path, implicit in page 4 of (Lipman et al., 2023)). $p_t(z_t|x) = \mathcal{N}(z_t|x \cdot \alpha_t, \beta_t^2 I_d)$.

**Fact C.3** (Marginal probability path, implicit in page 3 of (Lipman et al., 2023)). $p_t(z_t) = \sum_x p_t(z_t|x)p_{\mathrm{data}}(x)$.

Next, we introduce the flow matching, covering both the training phase and sampling phase. **Training.** After formulating the basic concepts in flow matching, we define its training loss function.

**Definition C.4** (First-order flow matching loss, implicit in page 3 on (Lipman et al., 2023)). *Let $t \in [0, 1]$ denote the timestep. The trajectory $z_t \in \mathbb{R}^d$ is defined in Definition 2.1 and the marginal velocity $v(z_t, t) \in \mathbb{R}^d$ is defined in Definition 2.4. To learn the marginal velocity fields, a neural network $u_\theta$ is trained to minimize the following loss function:*

$$L_{\mathrm{FO}}(\theta) := \mathop{\mathbb{E}}_{t, z_t} [\|u_\theta(z_t, t) - v(z_t, t)\|_2^2],$$

*where $t \sim \mathsf{Uniform}[0, 1]$ and $z_t \sim p_t(z_t)$.*

Directly computing the flow matching loss is challenging due to the marginalization in Definition 2.4. To overcome this, (Lipman et al., 2023) proposes the conditional flow matching loss as an alternative.

**Definition C.5** (Conditional first-order flow matching loss, implicit in page 4 on (Lipman et al., 2023)). *Let $t \in [0, 1]$ denote the timestep. The conditional velocity $v_t \in \mathbb{R}^d$ is defined in Definition 2.3. The neural network $u_\theta$ is trained to minimize the following loss function:*

$$L_{\mathrm{CFO}}(\theta) := \mathop{\mathbb{E}}_{t, x, \epsilon} [\|u_\theta(z_t, t) - v_t\|_2^2],$$

*where $t \sim \mathsf{Uniform}[0, 1]$, $x \sim p_{\mathrm{data}}(x)$, and $\epsilon \sim \mathcal{N}(\mu, \sigma^2 I_d)$.*

The effectiveness of the conditional flow matching loss is ensured by the following theorem.

**Theorem C.6** (Equivalence of loss functions, Theorem 2 on page 4 of (Lipman et al., 2023)). *Up to a constant independent of $\theta$, the loss functions $L_{\mathrm{FO}}(\theta)$ in Definition C.4 and $L_{\mathrm{CFO}}(\theta)$ in Definition C.5 are equivalent, i.e., $\nabla_\theta L_{\mathrm{FO}}(\theta) = \nabla_\theta L_{\mathrm{CFO}}(\theta)$. Here FO denotes first-order, and CFO denotes conditional first-order.*

Next, we present an equivalence result between the second-order flow matching loss and the conditional second-order flow matching loss.

**Fact C.7** (Equivalence of second-order flow matching loss). *Up to a constant independent of $\theta$, the second-order flow matching loss $L_{\mathrm{SO}}(\theta)$ in Definition 2.8 and the conditional second-order flow matching loss $L_{\mathrm{CSO}}(\theta)$ in Definition 2.9 are equivalent, i.e., $\nabla_\theta L_{\mathrm{SO}}(\theta) = \nabla_\theta L_{\mathrm{CSO}}(\theta)$.*

*Proof.* This directly follows from applying Theorem C.6 on both terms of $L_{\text{SO}}(\theta)$. □

**Sampling.** After training the flow matching model, we can sample random noise from the prior distribution and move it to the real data distribution with the learned velocity fields. This process can be formulated by the ODE and the corresponding initial value problem (IVP).

**Definition C.8** (First-order ODE, implicit in page 2 on (Lipman et al., 2023)). *Let $t \in [0,1]$ denote the timestep. The trajectory $z_t \in \mathbb{R}^d$ is defined in Definition 2.1, and the marginal velocity $v(z_t, t) \in \mathbb{R}^d$ is defined in Definition 2.4. Samples can be generated by solving the following ordinary differential equation (ODE) for $z_t$:*

$$\frac{\mathrm{d}z_t}{\mathrm{d}t} := v(z_t, t).$$

**Definition C.9** (First-order IVP). *Let $r, t \in [0,1]$ denote the two timesteps such that $r \leq t$. The trajectory $z_t \in \mathbb{R}^d$ is defined in Definition 2.1, and the marginal velocity $v(z_t, t) \in \mathbb{R}^d$ defined in Definition 2.4. Then, we define the first-order Initial Value Problem (IVP) as follows:*

$$z_r = z_t + \int_t^r v(z_\tau, \tau)\mathrm{d}\tau.$$

In practice, this integral in Definition C.9 is numerically approximated over discrete time steps. To solve it with the Euler solver, we have the following fact:

**Fact C.10** (Solving first-order MeanFlow IVP with $T$-step Euler solver). *Given $T \in \mathbb{Z}_+$ as the total iteration of sampling, and arbitrary $\{t_i\}_{i=1}^T$ that satisfy $t_i > t_{i+1}$ for all $i \in \{1, \ldots, T-1\}$ as timestep scheduler. Then, for all $i \in \{1, \ldots, T\}$, the first-order IVP defined as Definition C.9 can be solved iteratively with the $T$-step Euler solver as:*

$$z_{t_i} = z_{t_{i-1}} + (t_i - t_{i-1})v(z_{t_{i-1}}, t_i, t_{i-1}).$$

**Sampling.** The sampling process of the first-order MeanFlow model simply follows the same way as the iterative steps in Fact C.10, which is the same as the first-order flow matching. Since the average velocity satisfies the generalized consistency function, it allows a fast one-step sampling, i.e., $z_0 = z_1 - \overline{v}(z_1, 0, 1)$, where $z_1 \sim \mathcal{N}(\mu, \sigma^2 I_d)$ is sampled from the prior distribution.

To sample from second-order flow matching models, we need to consider a second-order ODE and its corresponding IVP. We first give the definition of second-order ODE.

**Definition C.11** (Second-order ODE). *Let $t \in [0,1]$ denote the timestep. The trajectory $z_t \in \mathbb{R}^d$, marginal velocity $v(z_t, t) \in \mathbb{R}^d$, and marginal acceleration $a(z_t, t) \in \mathbb{R}^d$ are defined in Definitions 2.1, 2.4, and 2.7, respectively. Samples can be generated by solving the following ODE:* $\frac{\mathrm{d}^2 z_t}{\mathrm{d}t^2} = a(z_t, t)$.

Then, we provide the definition of second-order IVP.

**Definition C.12** (Second-order IVP). *Let $t, r \in [0,1]$ denote two different timesteps such that $r \leq t$. The trajectory $z_t \in \mathbb{R}^d$, marginal velocity $v(z_t, t) \in \mathbb{R}^d$, and marginal acceleration $a(z_t, t) \in \mathbb{R}^d$ are defined in Definitions 2.1, 2.4, and 2.7, respectively. Then, we define the second-order IVP as follows:* $z_r := z_t + \int_t^r (v(z_{\tau_2}, \tau_2) + \int_t^{\tau_2} a(z_{\tau_1}, \tau_1)\mathrm{d}\tau_1)\mathrm{d}\tau_2$.

**Remark C.13.** *The first-order IVP in Definition C.9 is a special case of the second-order IVP above, since we can let $a(z_\tau, \tau) = 0$ for all $\tau \in [r, t]$ and exactly recover Definition C.9.*

**Fact C.14** (Solving second-order MeanFlow IVP with with $T$-step Euler solver). *Given $T \in \mathbb{Z}_+$ as the total iteration of sampling, and arbitrary $\{t_i\}_{i=1}^T$ that satisfy $t_i < t_{i+1}$ for all $i \in \{1, \ldots, T-1\}$ as timestep scheduler. Then, for all $i \in \{0, 1, \cdots, T\}$, the second-order IVP defined as Definition C.12 can be solved iteratively with the Euler solver as:* $z_{t_i} = z_{t_{i-1}} + (t_i - t_{i-1})v(z_{t_{i-1}}, t_i, t_{i-1}) + \frac{1}{2}(t_i - t_{i-1})^2 a(z_{t_{i-1}}, t_i, t_{i-1})$.

## C.2 CIRCUIT COMPLEXITY CLASS

This section defines Boolean circuits and basic facts for the analysis of circuit complexity. We begin with the fundamental definition of a single Boolean circuit.

**Definition C.15** (Boolean Circuit, implicit in page 102 on (Arora & Barak, 2009)). *Let $n \in \mathbb{Z}_+$. A Boolean circuit is a directed acyclic graph (DAG) that computes a function $C_n : \{0, 1\}^n \to \{0, 1\}$. The nodes of the graph are called gates. Nodes with an in-degree of 0 are input nodes, representing the $n$ Boolean variables. All other gates compute a Boolean function of their inputs, which are the outputs of preceding gates.*

Since a single circuit only handles a fixed input length, we use a family of circuits to recognize languages containing strings of any length.

**Definition C.16** (Circuit family recognizes languages, implicit in page 103 on (Arora & Barak, 2009)). *Let $L \subseteq \{0, 1\}^*$ be a language and let $C = \{C_n\}_{n \in \mathbb{N}}$ be a family of Boolean circuits. We say that $C$ recognizes $L$ if for every string $x \in \{0, 1\}^*$, the following holds: $C_{|x|}(x) = 1 \iff x \in L$.*

By placing constraints on the size and depth of these circuit families, we can define specific complexity classes, such as $\mathsf{NC}^i$.

**Definition C.17** ($\mathsf{NC}^i$,implicit in page 40 on (Arora & Barak, 2009)). *The class $\mathsf{NC}^i$ consists of languages decidable by families of Boolean circuits with polynomial size $O(\mathrm{poly}(n))$, depth $O((\log n)^i)$, and composed of AND, OR, and NOT gates with bounded fan-in.*

Allowing AND and OR gates to have unbounded fan-in enables circuits to recognize a broader class of languages. This leads to the definition of the class $\mathsf{AC}^i$.

**Definition C.18** ($\mathsf{AC}^i$, (Arora & Barak, 2009)). *The class $\mathsf{AC}^i$ is the set of languages recognizable by families of Boolean circuits with polynomial size $O(\mathrm{poly}(n))$ and depth $O((\log n)^i)$. These circuits use NOT, OR, and AND gates, where the OR and AND gates are permitted to have unbounded fan-in.*

Furthermore, since NOT, AND, and OR gates can be simulated by MAJORITY gates, we can define an even broader complexity class, $\mathsf{TC}^i$.

**Definition C.19** ($\mathsf{TC}^i$, (Arora & Barak, 2009)). *$\mathsf{TC}^i$ consists of languages recognized by Boolean circuits with depth $O((\log n)^i)$, size $O(\mathrm{poly}(n))$, and unbounded fan-in gates for NOT, OR, AND, and MAJORITY, where a MAJORITY gate outputs 1 when a majority of its inputs are active (1).*

**Remark C.20.** *The MAJORITY gates in the Definition C.19 of $\mathsf{TC}^i$ can be replaced by either MOD gates or THRESHOLD gates. A circuit containing any of these gate types is known as a threshold circuit.*

**Definition C.21** (P, implicit in page 27 on (Arora & Barak, 2009)). *The complexity class P is the set of all languages that can be decided by a deterministic Turing machine in polynomial time.*

**Fact C.22** (Hierarchy folklore, (Arora & Barak, 2009; Vollmer, 1999)). *The following inclusion relationships hold for all non-negative integers $i$: $\mathsf{NC}^i \subseteq \mathsf{AC}^i \subseteq \mathsf{TC}^i \subseteq \mathsf{NC}^{i+1} \subseteq \mathsf{P}$.*

**Definition C.23** ($L$-uniform, (Arora & Barak, 2009)). *A circuit family $C = \{C_n\}_{n \in \mathbb{N}}$ is $L$-uniform if a Turing machine exists that, on input $1^n$, generates a description of the circuit $C_n$ using $O(\log n)$ space. A language $L$ is in a class such as $L$-uniform $\mathsf{NC}^i$ if it is decided by an $L$-uniform circuit family $\{C_n\}$ that also meets the size and depth conditions of $\mathsf{NC}^i$.*

Next, we define a stricter form of uniformity based on a time constraint.

**Definition C.24** (DLOGTIME-uniform). *A circuit family $C = \{C_n\}_{n \in \mathbb{N}}$ is DLOGTIME-uniform if a Turing machine exists that, on input $1^n$, generates a description of the circuit $C_n$ in $O(\log n)$ time. A language is in a DLOGTIME-uniform class if it is decided by a DLOGTIME-uniform circuit family satisfying the corresponding resource constraints.*

We demonstrate several lemmas that define the basic operations' depth and width, which play fundamental roles in our circuit complexity analysis. First, we show that basic floating point operations can be implemented in $\mathsf{TC}^0$.

**Lemma C.25** (Operations on floating point numbers in $\mathsf{TC}^0$, Lemma 10 and Lemma 11 of (Chiang, 2025)). *Assume the precision $p \leq \mathrm{poly}(n)$. Then we have:*

- *Part 1. Given two $p$-bits float point numbers $x_1$ and $x_2$. Let the addition, division, and multiplication operations of $x_1$ and $x_2$ be outlined in (Chiang, 2025). Then, these operations can be simulated by a size bounded by $\mathrm{poly}(n)$ and constant depth bounded by $d_{\mathrm{std}}$ DLOGTIME-uniform threshold circuit.*

- *Part 2. Given $n$ $p$-bits float point number $x_1, \ldots, x_n$. The iterated multiplication of $x_1, x_2 \ldots, x_n$ can be simulated by a size bounded by $\mathrm{poly}(n)$ and constant depth bounded by $d_{\otimes}$ DLOGTIME-uniform threshold circuit.*

- *Part 3. Given $n$ $p$-bits float point number $x_1, \ldots, x_n$. The iterated addition of $x_1, x_2 \ldots, x_n$ can be simulated by a size bounded by $\mathrm{poly}(n)$ and constant depth bounded by $d_{\oplus}$ DLOGTIME-uniform threshold circuit. To be noticed, there is a rounding operation after the the summation is completed.*

Next, we show that the exponential function can also be approximated in $\mathsf{TC}^0$.

**Lemma C.26** (Approximating the Exponential Operation in $\mathsf{TC}^0$, Lemma 12 of (Chiang, 2025))**.** *Assume the precision $p \leq \mathrm{poly}(n)$. Given any number $x$ with $p$-bit float point, the $\exp(x)$ function can be approximated by a uniform threshold circuit. This circuit has a size bounded by $\mathrm{poly}(n)$ and a constant depth $d_{\mathrm{exp}}$, and it guarantees a relative error of at most $2^{-p}$.*

Last, we demonstrate that the square root function can be approximated in $\mathsf{TC}^0$.

**Lemma C.27** (Approximating the Square Root Operation in $\mathsf{TC}^0$, Lemma 12 of (Chiang, 2025))**.** *Assume the precision $p \leq \mathrm{poly}(n)$. Given any number $x$ with $p$-bit float point, the $\sqrt{x}$ function can be approximated by a uniform threshold circuit. This circuit has a size bounded by $\mathrm{poly}(n)$ and a constant depth $d_{\mathrm{sqrt}}$, and it guarantees a relative error of at most $2^{-p}$.*

## C.3 ViT Model Formation

In order to analyze the circuit complexity of MeanFlow, we first provide the formal formulation for Vision Transformer (ViT) (Dosovitskiy et al., 2021), which is an essential component in real-world MeanFlow implementation. First, we show the definition of the attention matrix.

**Definition C.28** (Attention matrix)**.** *We use $W_Q, W_K \in \mathbb{F}_p^{d \times d}$ to denote the weight matrix of the query and key. Let $X \in \mathbb{F}_p^{n \times d}$ represent the input of the attention layer. Then, we use $A \in \mathbb{F}_p^{n \times n}$ to denote the attention matrix. Specifically, we denote the element of the attention matrix as the following: $A_{i,j} := \exp(X_{i,*} W_Q W_K^\top X_{j,*}^\top)$.*

Next, we demonstrate the definition of the single attention layer.

**Definition C.29** (Single attention layer)**.** *We use $W_V \in \mathbb{F}_p^{d \times d}$ to denote the weight matrix of value. Let $X \in \mathbb{F}_p^{n \times d}$ represent the input of the attention layer. Let $A$ denote the attention matrix defined in Definition C.28. Let $D := \mathrm{diag}(A \mathbf{1}_n)$ denote a size $n \times n$ matrix. Then, we use $\mathsf{Attn}$ to denote the attention layer. Specifically, we have $\mathsf{Attn}(X) := D^{-1} A X W_V$.*

Similarly, we define the multi-layer perceptron layer.

**Definition C.30** (Multi-layer perceptron (MLP) layer)**.** *Given an input matrix $X \in \mathbb{F}_p^{n \times d}$. Let $i \in [n]$. We use $g^{\mathrm{MLP}}$ to denote the MLP layer. Specifically, we have $g^{\mathrm{MLP}}(X)_{i,*} := W \cdot X_{i,*} + b$.*

Furthermore, the definition of the layer-wise normalization layer is as follows.

**Definition C.31** (Layer-wise normalization (LN) layer)**.** *Given an input matrix $X \in \mathbb{F}_p^{n \times d}$. Let $i \in [n]$. We use $g^{\mathrm{LN}}$ to denote the LN layer. Specifically, we have $g^{\mathrm{LN}}(X)_{i,*} := \frac{X_{i,*} - \mu_i}{\sqrt{\sigma_i^2}}$, where $\mu_i := \sum_{j=1}^d X_{i,j}/d$, and $\sigma_i^2 := \sum_{j=1}^d (X_{i,j} - \mu_i)^2/d$.*

By combining these definitions, we can define the Vision Transformer.

**Definition C.32** (Vision Transformer (ViT))**.** *Assume the ViT has $m$ transformer layers. At the $i$-th layer, let $\mathsf{Attn}_i$ denote the self-attention layer (Definition C.29), $\mathsf{LN}_{i,1}, \mathsf{LN}_{i,2}$ denote the two layer normalization layers (Definition C.31), and $\mathsf{MLP}_i$ denote the MLP layer (Definition C.30).*

*Let $X_0 \in \mathbb{F}_p^{n \times d}$ be the input. Then, for each $i \in [m]$, define: $Y_i := \mathsf{Attn}_i \circ \mathsf{LN}_{i,1}(X_{i-1}) + X_{i-1} \in \mathbb{F}_p^{n \times d}, X_i := \mathsf{MLP}_i \circ \mathsf{LN}_{i,2}(Y_i) + Y_i \in \mathbb{F}_p^{n \times d}$, where $\circ$ denotes function composition. We use $\mathsf{ViT}(X_0) := X_m$ to denote the computation of ViT as a function.*

## D  FEASIBILITY OF SECOND-ORDER MeanFlow

This section supplies the proofs omitted from Section 3. In Section D.1, we prove the boundary conditions for both average velocity and average acceleration. In Section D.2, we prove the consistency constraints for both average velocity and average acceleration. In Section D.3, we prove the equivalence of the MeanFlow loss functions. In Section D.4, we demonstrate the effectiveness of the MeanFlow and Second-order MeanFlow loss function.

### D.1  BOUNDARY CONDITIONS

In this section, we establish the boundary conditions of average velocity and acceleration. We begin with the boundary condition for average velocity, which connects it to the marginal velocity.

**Lemma D.1** (Boundary condition of average velocity, formal version of Lemma 3.5). *The relationship between the average velocity $\overline{v}$ (Definition 3.1) and the marginal velocity $v$ (Definition 2.4) is established by the boundary condition in Definition 3.3 as follows:*

$$\overline{v}(z_r, r, r) = v(z_r, r),$$

*and*

$$\overline{v}(z_t, t, t) = v(z_t, t).$$

*Proof.* **Part 1:** $\overline{v}(z_r, r, r) = v(z_r, r)$**.** By Definition 3.1, the average velocity is given by:

$$\overline{v}(z_t, r, t) = \frac{1}{t-r} \int_r^t v(z_\tau, \tau) d\tau.$$

We can view this as the mean value of the function $v$ over the interval $[r, t]$. As the interval length $t - r$ approaches 0, this mean value converges to the value of the integrand at that point. Thus, we have:

$$\lim_{t \to r} \overline{v}(z_t, r, t) = \lim_{t \to r} \frac{1}{t-r} \int_r^t v(z_\tau, \tau) d\tau$$
$$= v(z_r, r),$$

where the first step follows from the definition of $\overline{v}$ in Definition 3.1, and the second step follows from the fundamental theorem of calculus.

**Part 2:** $\overline{v}(z_t, t, t) = v(z_t, t)$**.** According to Definition 3.1, the average velocity $\overline{v}$ is defined as the mean value of the velocity function v over the interval $[r, t]$:

$$\overline{v}(z_t, r, t) = \frac{1}{t-r} \int_r^t v(z_\tau, \tau) d\tau.$$

As the interval length $(t - r)$ shrinks to zero, this mean value converges to the value of the integrand at that point. We can show this formally by taking the limit as $r \to t$. By the Fundamental Theorem of Calculus, we have:

$$\lim_{r \to t} \overline{v}(z_t, r, t) = \lim_{r \to t} \frac{1}{t-r} \int_r^t v(z_\tau, \tau) d\tau$$
$$= v(z_t, t).$$

The first step applies the definition of $\overline{v}$, and the second is a direct result of the theorem. This demonstrates that the average velocity $\overline{v}$ converges to the instantaneous (or marginal) velocity v in the limit.

Thus, we prove that the average velocity $\overline{v}$ equals the marginal velocity $v$ at the limit. $\qquad\square$

We now apply the same logic to acceleration. The following formalizes how the average acceleration converges to the marginal acceleration.

**Lemma D.2** (Boundary conditions of average acceleration, formal version of Lemma 3.17). *Given the average acceleration $\overline{a}$ defined in Definition 3.16 and the marginal acceleration $a$ defined in Definition 2.7, $\overline{a}$ converges to $a$ as the time interval shrinks to zero. Specifically:*

$$\overline{a}(z_r, r, r) = a(z_r, r)$$

*and*

$$\overline{a}(z_t, t, t) = a(z_t, t).$$

*Proof.* **Part 1:** $\overline{a}(z_t, t, t) = a(z_t, t)$**.** By Definition 3.16, the average acceleration over the interval $[r, t]$ is:

$$\overline{a}(z_t, r, t) = \frac{1}{t - r} \int_r^t a(z_\tau, \tau) d\tau.$$

Taking the limit as $r \to t$, we get:

$$\lim_{r \to t} \overline{a}(z_t, r, t) = \lim_{r \to t} \frac{\int_r^t a(z_\tau, \tau) d\tau}{t - r}$$
$$= a(z_t, t),$$

where the first step follows from the definition of $\overline{a}$ in Definition 3.16, and the second step follows from the fundamental theorem of calculus.

**Part 2:** $\overline{a}(z_r, r, r) = a(z_r, r)$**.** The proof is symmetrical to Part 1. We take the limit of the average acceleration over the interval $[r, t]$ as $t$ approaches $r$. By the same application of the Fundamental Theorem of Calculus, this limit resolves to the instantaneous acceleration at time $r$:

$$\lim_{t \to r} \overline{a}(z_t, r, t) = \lim_{t \to r} \frac{\int_r^t a(z_\tau, \tau) d\tau}{t - r}$$
$$= a(z_r, r).$$

Thus, we complete the proof. $\square$

### D.2 CONSISTENCY CONSTRAINT

In this section, we give the consistency constraints for average velocity and acceleration. We begin by demonstrating that average velocity (Definition 3.1) satisfies the consistency constraint.

**Lemma D.3** (Consistency constraint of average velocity, formal version of Lemma 3.6). *Let $r, t$ denote two different timesteps. The timestep $s$ is any intermediate time between $r$ and $t$. The marginal velocity $v$ is defined in Definition 2.4. The average velocity $\overline{v}$ defined in Definition 3.1 satisfies an additive consistency constraint:*

$$(t - r)\overline{v}(z_t, r, t) = (s - r)\overline{v}(z_s, r, s) + (t - s)\overline{v}(z_t, s, t).$$

*Proof.* For the marginal velocity $v$ integrated over time $\tau$, we have:

$$\int_r^t v(z_\tau, \tau) d\tau = \int_r^s v(z_\tau, \tau) d\tau + \int_s^t v(z_\tau, \tau) d\tau,$$

where the first step follows from the additivity of the integral.

From the definition of average velocity in Definition 3.1, $\overline{v}(z_t, r, t) = \frac{1}{t-r} \int_r^t v d\tau$, we can express the displacement (the integral) as $(t - r)\overline{v}(z_t, r, t)$. By substituting this relationship into the integral additivity equation above, we arrive at the consistency constraint:

$$(t - r)\overline{v}(z_t, r, t) = (s - r)\overline{v}(z_s, r, s) + (t - s)\overline{v}(z_t, s, t).$$

This demonstrates that $\overline{v}$ satisfies its inherent consistency constraint.

$\square$

Next, we show that the consistency constraint applies to average acceleration (Definition 3.16).

**Lemma D.4** (Consistency constraint of average acceleration, formal version of Lemma 3.18)**.** *Let $r, s,$ and $t$ be three distinct time steps, such that $s$ lies between $r$ and $t$. Given the marginal acceleration $a$ (as defined in Definition 2.7), the average acceleration $\overline{a}$ (as defined in Definition 3.16) satisfies the following additive consistency constraint:*

$$(t - r)\overline{a}(z_t, r, t) = (s - r)\overline{a}(z_s, r, s) + (t - s)\overline{a}(z_t, s, t).$$

*Proof.* The integral of marginal acceleration $a(z_\tau, \tau)$ over time $\tau$ exhibits additivity:

$$\int_r^t a(z_\tau, \tau)d\tau = \int_r^s a(z_\tau, \tau)d\tau + \int_s^t a(z_\tau, \tau)d\tau.$$

From Definition 3.16, the average acceleration $\overline{a}(z_t, r, t)$ is defined as $\frac{1}{t-r}\int_r^t a(z_\tau, \tau)d\tau$. This allows us to express the integral of acceleration over a given time interval as the product of the time interval and the average acceleration over that interval. Specifically, $\int_r^t a(z_\tau, \tau)d\tau = (t-r)\overline{a}(z_t, r, t)$.

Substituting this relationship into the additive integral equation yields the consistency constraint:

$$(t - r)\overline{a}(z_t, r, t) = (s - r)\overline{a}(z_s, r, s) + (t - s)\overline{a}(z_t, s, t).$$

Thus, we complete the proof. $\qquad\square$

### D.3 EQUIVALENCE OF MeanFlow LOSS FUNCTIONS

In this section, we show the equivalence between the practial and ideal loss functions for first-order MeanFlow.

**Theorem D.5** (Equivalence of MeanFlow loss functions, formal version of Theorem 3.13)**.** *The loss function $L_{\text{FOM}}(\theta)$ in Definition 3.11 is equivalent to the ideal loss function $L_{\text{FOM}}^*(\theta)$ in Definition 3.8. Here FOM denotes first-order MeanFlow.*

*Proof.* We prove the equivalence of Eq. (1) and Eq. (2) from both sides.

**Part 1: Eq. (1) $\implies$ Eq. (2).** The definition of Eq. (1) is as follows:

$$\overline{v}(z_t, r, t) = \frac{1}{t - r}\int_r^t v(z_\tau, \tau)\mathrm{d}\tau.$$

We rewrite Eq. (1) as:

$$(t - r)\overline{v}(z_t, r, t) = \int_r^t v(z_\tau, \tau)\mathrm{d}\tau. \tag{4}$$

Next, we differentiate both sides of Eq. (4) with respect to $t$, treating $r$ as independent of $t$. This yields:

$$\frac{\mathrm{d}}{\mathrm{d}t}(t - r)\overline{v}(z_t, r, t) = \frac{\mathrm{d}}{\mathrm{d}t}\int_r^t v(z_\tau, \tau)\mathrm{d}\tau.$$

Applying the product rule to the left-hand side and the fundamental theorem of calculus to the right-hand side, we obtain:

$$\overline{v}(z_t, r, t) + (t - r)\frac{\mathrm{d}}{\mathrm{d}t}\overline{v}(z_t, r, t) = v(z_\tau, \tau),$$

Rearranging the terms, we obtaion Eq. (2):

$$\overline{v}(z_t, r, t) = v(z_t, t) - (t - r)\frac{\mathrm{d}}{\mathrm{d}t}\overline{v}(z_t, r, t).$$

**Part 2: Eq. (2) $\implies$ Eq. (1).** The definition of Eq. (2) is as follows:

$$\overline{v}(z_t, r, t) = v(z_t, t) - (t - r)\frac{\mathrm{d}}{\mathrm{d}t}\overline{v}(z_t, r, t).$$

We rewrite Eq. (2) as follows:

$$v(z_t, t) = \overline{v}(z_t, r, t) + (t - r)\frac{\mathrm{d}}{\mathrm{d}t}\overline{v}(z_t, r, t) \tag{5}$$

$$= \overline{v}(z_t, r, t)\frac{\mathrm{d}}{\mathrm{d}t}(t - r) + (t - r)\frac{\mathrm{d}}{\mathrm{d}t}\overline{v}(z_t, r, t)$$

$$= \frac{\mathrm{d}}{\mathrm{d}t}(t - r)\overline{v}(z_t, r, t),$$

where the first step follows from the definition of Eq. (2), the second step follows from the product rule of differentiation, and the third step follows from $\frac{\mathrm{d}}{\mathrm{d}t}(t - r) = 1$.

In general, equality of derivatives does not imply equality of integrals: they may differ by a constant. Therefore, based on the third step of Equation (5), the following equation can be directly implied:

$$(t - r)\overline{v}(z_t, r, t) + C_1 = \int_r^t v(z_\tau, \tau)d\tau + C_2. \tag{6}$$

Choosing $t = r$ in Eq. (6), we have $C_1 = C_2$. Substituting $C_1 = C_2$ into Eq (6), we get the following equation:

$$(t - r)\overline{v}(z_t, r, t) = \int_r^t v(z_\tau, \tau)d\tau.$$

We rewrite the equation to obtain Eq. (1) as follows:

$$\overline{v}(z_t, r, t) = \frac{1}{(t - r)}\int_r^t v(z_\tau, \tau)d\tau.$$

Thus, we complete the proof.

$\square$

### D.4 EFFECTIVENESS OF MeanFlow AND SECOND-ORDER MeanFlow LOSS FUNCTION

In this section, we present the effectiveness of MeanFlow and second-order MeanFlow loss function. First, we show that the first-order MeanFlow loss function can be computed efficiently.

**Theorem D.6** (Efficiency of MeanFlow loss, formal version of Theorem 3.14)**.** *The loss function $L_{\mathrm{FOM}}(\theta)$ in Definition 3.11 can be evaluated efficiently with the conditional velocity $v_t$ in Definition 2.3 and Jacobian-Vector Products (JVPs) of the neural network $u_{1,\theta_1}$, i.e.,*

$$\overline{v}(z_t, r, t) = v_t - (t - r)(v_t\partial_z u_\theta + \partial_t u_\theta).$$

*Proof.* We fist compute the total derivative of $\overline{v}(z_t, r, t)$ as follows:

$$\frac{\mathrm{d}}{\mathrm{d}t}\overline{v}(z_t, r, t) = \frac{\mathrm{d}z_t}{\mathrm{d}t}\partial_{z_t}u + \frac{\mathrm{d}r}{\mathrm{d}t}\partial_r u + \frac{\mathrm{d}t}{\mathrm{d}t}\partial_t u \tag{7}$$

$$= v(z_t, t)\partial_{z_t}u + \partial_t u,$$

where the first step follows from the chain rule and the second step follows from $\frac{\mathrm{d}z_t}{\mathrm{d}t} = v(z_t, t)$ in Definition C.8, $\frac{\mathrm{d}r}{\mathrm{d}t} = 0$ and $\frac{\mathrm{d}t}{\mathrm{d}t} = 1$.

Besides, we also have

$$\overline{v}(z_t, r, t) = \frac{1}{t - r}\int_r^t v(z_\tau, \tau)\mathrm{d}\tau$$

$$= v(z_t, t) - (t - r)\frac{\mathrm{d}}{\mathrm{d}t}\overline{v}(z_t, r, t)$$

$$= v(z_t, t) - (t - r)(v(z_t, t)\partial_{z_t}\overline{v} + \partial_t\overline{v}), \tag{8}$$

where the first step follows from definition of $\overline{v}(z_t, r, t)$ in Definition 3.1, the second step follows from Theorem 3.13, and the third step follows from Eq. (7).

The velocity $v(z_t, t)$ given in Eq. (8) is the marginal velocity. Following (Lipman et al., 2023), we replace this with the conditional velocity. Then, the objective function is:

$$\overline{v}(z_t, r, t) = v_t - (t - r)(v_t\partial_z u_\theta + \partial_t u_\theta).$$

This finishes the proof. □

Next, we establish the equivalence between the practical and ideal second-order loss functions.

**Theorem D.7** (Effectivenss of second-order MeanFlow, formal version of Theorem 3.22)**.** *The loss function $L_{\mathrm{SOM}}(\theta)$ in Definition 3.21 is equivalent to the ideal loss function $L_{\mathrm{SOM}}^*(\theta)$ in Definition 3.20. Here* SOM *denotes second-order* MeanFlow.

*Proof.* The key to showing the effectiveness of the second-order MeanFlow is showing the equivalence between Eq. (3) of Definition 3.16 and the following equation:

$$\overline{a}(z_t, r, t) = a(z_t, t) - (t - r)\frac{\mathrm{d}}{\mathrm{d}t}\overline{a}(z_t, r, t). \tag{9}$$

We prove the equivalence of the integral form Eq. (3) and the differential form Eq. (9) in two parts.

**Part 1: Eq. (3) $\implies$ Eq. (9)** The integral definition of the average acceleration is:

$$\overline{a}(z_t, r, t) = \frac{1}{t - r}\int_r^t a(z_\tau, \tau)\mathrm{d}\tau.$$

Multiplying by $(t - r)$, we get:

$$(t - r)\overline{a}(z_t, r, t) = \int_r^t a(z_\tau, \tau)\mathrm{d}\tau. \tag{10}$$

Next, we differentiate both sides of Eq. (10) with respect to $t$, treating $r$ as a constant:

$$\frac{\mathrm{d}}{\mathrm{d}t}\left[(t - r)\overline{a}(z_t, r, t)\right] = \frac{\mathrm{d}}{\mathrm{d}t}\int_r^t a(z_\tau, \tau)\mathrm{d}\tau.$$

Applying the product rule to the left-hand side and the Fundamental Theorem of Calculus to the right-hand side yields:

$$\overline{a}(z_t, r, t) + (t - r)\frac{\mathrm{d}}{\mathrm{d}t}\overline{a}(z_t, r, t) = a(z_t, t).$$

Rearranging the terms, we obtain the differential form in Eq. (9):

$$\overline{a}(z_t, r, t) = a(z_t, t) - (t - r)\frac{\mathrm{d}}{\mathrm{d}t}\overline{a}(z_t, r, t).$$

**Part 2: Eq. (9) $\implies$ Eq. (3)** We start with the differential form from Eq. (9):

$$\overline{a}(z_t, r, t) = a(z_t, t) - (t - r)\frac{\mathrm{d}}{\mathrm{d}t}\overline{a}(z_t, r, t).$$

Rearranging the equation to isolate $a(z_t, t)$, we find:

$$a(z_t, t) = \overline{a}(z_t, r, t) + (t - r)\frac{\mathrm{d}}{\mathrm{d}t}\overline{a}(z_t, r, t).$$

We recognize the right-hand side as the result of the product rule for differentiation applied to $(t - r)\overline{a}(z_t, r, t)$:

$$a(z_t, t) = \frac{\mathrm{d}}{\mathrm{d}t}\left[(t - r)\overline{a}(z_t, r, t)\right].$$

Now, we integrate both sides with respect to a dummy variable $\tau$ from $r$ to $t$:

$$\int_r^t a(z_\tau, \tau)\mathrm{d}\tau = \int_r^t \frac{\mathrm{d}}{\mathrm{d}\tau}\left[(\tau - r)\overline{a}(z_\tau, r, \tau)\right]\mathrm{d}\tau.$$

Applying the Fundamental Theorem of Calculus, we have:

$$\int_r^t a(z_\tau, \tau)\mathrm{d}\tau = (t - r)\overline{a}(z_t, r, t)$$

Finally, dividing by $(t - r)$ for $t \neq r$ yields the integral definition from Eq. (3):

$$\overline{a}(z_t, r, t) = \frac{1}{t - r}\int_r^t a(z_\tau, \tau)\mathrm{d}\tau.$$

Therefore, applying Theorem C.6, we finish the proof of equivalence.

$\square$

Finally, we demonstrate that the second-order loss function is also computed efficiently.

**Theorem D.8** (Efficiency of second-order MeanFlow, formal version of Theorem 3.23)**.** *The second-order* MeanFlow *loss function* $L_{\mathrm{SOM}}(\theta)$ *in Definition 3.21 can be evaluated efficiently with the conditional velocity* $v_t$ *and conditional acceleration* $a_t$ *in and Jacobian-Vector Products (JVPs) of the neural networks* $u_{1,\theta_1}$ *and* $u_{2,\theta_2}$, *i.e.*,

$$\overline{v}(z_t, r, t) = v_t - (t - r)(v_t \partial_z u_{1,\theta_1} + \partial_t u_{1,\theta_1}),$$
$$\overline{a}(z_t, r, t) = a_t - (t - r)(a_t \partial_z u_{2,\theta_2} + \partial_t u_{2,\theta_2}).$$

*Proof.* We first the compute the total derivative of $\overline{a}(z_t, r, t)$ as follows:

$$\frac{\mathrm{d}}{\mathrm{d}t}\overline{a}(z_t, r, t) = \frac{\mathrm{d}z_t}{\mathrm{d}t}\partial_{z_t}u + \frac{\mathrm{d}r}{\mathrm{d}t}\partial_r u + \frac{\mathrm{d}t}{\mathrm{d}t}\partial_t u \qquad (11)$$
$$= v(z_t, t)\partial_{z_t}u + \partial_t u,$$

where the first step is due to the chain rule and the second step is obtained by $\frac{\mathrm{d}z_t}{\mathrm{d}t} = v(z_t, t)$ in Definition C.8, $\frac{\mathrm{d}r}{\mathrm{d}t} = 0$ and $\frac{\mathrm{d}t}{\mathrm{d}t} = 1$.

Next, we have:

$$\overline{a}(z_t, r, t) = \frac{1}{t - r}\int_r^t v(z_\tau, \tau)\mathrm{d}\tau$$

$$= a(z_t, t) - (t - r)\frac{\mathrm{d}}{\mathrm{d}t}\overline{a}(z_t, r, t)$$
$$= a(z_t, t) - (t - r)(v(z_t, t)\partial_{z_t}\overline{a} + \partial_t\overline{a}),$$

where the first step is due to definition of $\overline{a}(z_t, r, t)$ in Definition 3.16, the second step is obtained by Theorem 3.22, and the third step follows from Eq. (11).

This finishes the proof. $\square$

# E  CIRCUIT COMPLEXITY OF SECOND-ORDER MeanFlow

This section provides the missing proofs for Section 4. In Section E.1,we prove that ViT computation belongs to $\mathsf{TC}^0$. In Section E.2, we show that sampling with a T-step Euler solver for both MeanFlow and second-order MeanFlow also belongs to $\mathsf{TC}^0$.

### E.1 CIRCUIT COMPLEXITY OF VIT

In this section, we provide the circuit complexity of ViT.

We first introduce a the circuit complexity class of matrix multiplication, which is a crucial component for our work.

**Lemma E.1** (Matrix Multiplication belongs to $\mathsf{TC}^0$ class, Lemma 4.2 in (Chen et al., 2024a)). *Let $M \in \mathbb{F}_p^{n_1 \times d}$ and $M \in \mathbb{F}_p^{d \times n_2}$. If precision $p \leq \mathrm{poly}(n)$, $n_1, n_2 \leq \mathrm{poly}(n)$, and $d \leq n$. Then $MN$ can be computed by a uniform threshold circuit of depth $(d_{\mathrm{std}} + d_\oplus)$ and size $\mathrm{poly}(n)$.*

Then, we demonstrate that the computation of attention matrix belongs to $\mathsf{TC}^0$ class.

**Lemma E.2** (Attention matrix computation belongs to $\mathsf{TC}^0$ class, Lemma 5.3 in (Ke et al., 2025)). *Assume the precision $p \leq \mathrm{poly}(n)$, then we can use a size bounded by $\mathrm{poly}(n)$ and constant depth $3(d_{\mathrm{std}} + d_\oplus) + d_{\exp}$ uniform threshold circuit to compute the attention matrix $A$ defined in Definition C.28.*

Next, we outline the circuit complexity of the computation of MLP layers.

**Lemma E.3** (MLP computation falls within $\mathsf{TC}^0$ class, Lemma 4.5 of (Chen et al., 2024a)). *Assume the precision $p \leq \mathrm{poly}(n)$. Then, we can use a size bounded by $\mathrm{poly}(n)$ and constant depth $2d_{\mathrm{std}} + d_\oplus$ uniform threshold circuit to simulate the MLP layer in Definition C.30.*

In this work, LayerNorm is also a key component, which is used multiple times.

**Lemma E.4** (LN computation falls within $\mathsf{TC}^0$ class, Lemma 4.6 of (Chen et al., 2024a)). *Assume the precision $p \leq \mathrm{poly}(n)$, then we can use a size bounded by $\mathrm{poly}(n)$ and constant depth $5d_{\mathrm{std}} + 2d_\oplus + d_{\mathrm{sqrt}}$ uniform threshold circuit to simulate the Layer-wise Normalization layer defined in Definition C.31.*

Next, we give the circuit complexity for computing the value $Y_i$.

**Lemma E.5** ($Y_i$ of ViT computation in $\mathsf{TC}^0$). *Assume the precision $p \leq \mathrm{poly}(n)$, then for a $m$ layer ViT, we can use a size bounded by $\mathrm{poly}(n)$ and constant depth $m(9d_{\mathrm{std}} + 5d_\oplus + d_{\mathrm{sqrt}} + d_{\exp})$ uniform threshold circuit to simulate the computation of $Y_i$ defined in Definition C.32.*

*Proof.* By using the result of Lemma E.4, we can apply a size bounded by $\mathrm{poly}(n)$ and a constant depth $5d_{\mathrm{std}} + 2d_\oplus + d_{\mathrm{sqrt}}$ uniform threshold circuit to simulate the computation of $\mathsf{LN}_{i,1}$ layer, defined in Definition C.31, for each $i \in [m]$.

By using the result of Lemma E.2, we can apply a size bounded by $\mathrm{poly}(n)$ and a constant depth $3(d_{\mathrm{std}} + d_\oplus) + d_{\exp}$ uniform threshold circuit to simulate $\mathsf{Attn}_i$ defined in Definition C.29, for each $i \in [m]$

For the residual addition, since the matrices have dimensions $n \times d$, we can use $nd \leq O(n^2)$ parallel adders. The overall circuit has depth $d_{\mathrm{std}}$ and size $\mathrm{poly}(n)$. Therefore, for each $i \in [m]$, we require a circuit of size $\mathrm{poly}(n)$ and depth $d_{\mathrm{std}}$.

Then, we can have that the size of the uniform threshold circuit is bounded by $\mathrm{poly}(n)$, and the total depth of the circuit is $m(9d_{\mathrm{std}} + 5d_\oplus + d_{\mathrm{sqrt}} + d_{\exp})$. □

Next, we give the circuit complexity for computing the value $X_i$ in the MLP block.

**Lemma E.6** ($X_i$ of ViT computation in $\mathsf{TC}^0$). *Assume the precision $p \leq \mathrm{poly}(n)$, then for a $m$ layer ViT, we can use a size bounded by $\mathrm{poly}(n)$ and constant depth $m(8d_{\mathrm{std}} + 3d_\oplus + d_{\mathrm{sqrt}})$ uniform threshold circuit to simulate the computation of $X_i$ defined in Definition C.32.*

*Proof.* By using the result of Lemma E.4, we can apply a size bounded by $\mathrm{poly}(n)$ and a constant depth $5d_{\mathrm{std}} + 2d_\oplus + d_{\mathrm{sqrt}}$ uniform threshold circuit to simulate the computation of $\mathsf{LN}_{i,2}$ layer, defined in Definition C.31, for each $i \in [m]$.

By using the result of Lemma E.3, we can apply a size bounded by $\mathrm{poly}(n)$ and a constant depth $2d_{\mathrm{std}} + d_\oplus$ uniform threshold circuit to simulate $\mathsf{MLP}_i$ defined in Definition C.30, for each $i \in [m]$

For the residual addition, since the matrices have dimensions $n \times d$, we can use $n(d + 2)$ parallel adders. The overall circuit has depth $d_{\text{std}}$ and size $\text{poly}(n)$. Therefore, for each $i \in [m]$, we require a circuit with size $\text{poly}(n)$ and depth $d_{\text{std}}$.

Then, we can have that the size of the uniform threshold circuit is bounded by $\text{poly}(n)$, and the total depth of the circuit is $m(8d_{\text{std}} + 3d_{\oplus} + d_{\text{sqrt}})$. $\qquad\square$

Finally, we combine these lemmas to show the ViT can be simulated by a $\mathsf{TC}^0$ circuit.

**Lemma E.7** (ViT computation in $\mathsf{TC}^0$). *Assume the number of transformer layers $m = O(1)$. Assume the precision $p \leq \text{poly}(n)$. Subsequently, we can apply a uniform threshold circuit to simulate the ViT defined in Definition C.32. The circuit has size $\text{poly}(n)$ and $O(1)$ depth.*

*Proof.* Applying Lemma E.5, we can apply a size bounded by $\text{poly}(n)$ and a constant depth $m(9d_{\text{std}} + 5d_{\oplus} + d_{\text{sqrt}} + d_{\exp})$ uniform threshold circuit to simulate the computation of $Y_i$ defined in Definition C.32.

By using the result of Lemma E.6, we can apply a size bounded by $\text{poly}(n)$ and a constant depth $m(8d_{\text{std}} + 3d_{\oplus} + d_{\text{sqrt}})$ uniform threshold circuit to simulate the computation of $X_i$ defined in Definition C.32.

Thus, we can have that the size of the uniform threshold circuit is bounded by $\text{poly}(n)$, and the total depth of the circuit is $m(17d_{\text{std}} + 8d_{\oplus} + 2d_{\text{sqrt}} + d_{\exp})$, since $m = O(1)$, thus we have $O(1)$ total depth. $\qquad\square$

### E.2 Circuit Complexity of MeanFlow with Euler Solver

In this section, we show the circuit complexity of MeanFlow with Euler Solver. We begin by establishing the circuit complexity for the sampling process of the first-order MeanFlow model.

**Theorem E.8** (MeanFlow sampling with $T$-step Euler solver belongs to $\mathsf{TC}^0$, formal version of Theorem 4.3). *Given $T \in \mathbb{Z}_+$ as the total iteration of sampling, and arbitrary $\{t_i\}_{i=1}^{T}$ that satisfy $t_i > t_{i+1}$ for all $i \in \{1, \ldots, T - 1\}$ as timestep scheduler. Assume the precision $p \leq \text{poly}(n)$, the number of transformer layers $m = O(1)$, and $T = O(1)$. Then we can use a size bounded by $\text{poly}(n)$ and constant depth $T(m(17d_{\text{std}} + 8d_{\oplus} + 2d_{\text{sqrt}} + d_{\exp}) + 5d_{\text{std}})$ uniform threshold circuit to simulate the MeanFlow $T$-step sampling defined in Definition 4.1.*

*Proof.* The operation $t_i \mathbf{1}_n$ and $t_{i-1} \mathbf{1}_n$ can be implemented by a uniform threshold circuit of size $n < \text{poly}(n)$ and depth $d_{\text{std}}$.

Using Lemma E.7, we can simulate the ViT (Definition C.32) with a uniform threshold circuit. This circuit has a size bounded by $\text{poly}(n)$ and a constant depth $m(17d_{\text{std}} + 8d_{\oplus} + 2d_{\text{sqrt}} + d_{\exp})$.

The $t_i - t_{i-1}$ operation can be simulated with a uniform threshold circuit with size of $1 < \text{poly}(n)$ and depth of $d_{\text{std}}$.

The multiplication between result of $t_i - t_{i-1}$ and result of sliced ViT can be implemented by a uniform threshold circuit of depth $d_{\text{std}}$ and size $nd < \text{poly}(n)$.

The addition between the result from previous step and $Z_{t_{i-1}}$ can be implemented by a uniform threshold circuit of depth $d_{\text{std}}$ and size $nd < \text{poly}(n)$.

Since we have total $T$ iterations, we can have that the size of the uniform threshold circuit is bounded by $\text{poly}(n)$, and the total depth of the circuit is $T(m(17d_{\text{std}} + 8d_{\oplus} + 2d_{\text{sqrt}} + d_{\exp}) + 5d_{\text{std}}) = O(1)$ total depth since $T = O(1)$ and $m = O(1)$. $\qquad\square$

Then, we extend this analysis to the sampling process for the second-order MeanFlow model.

**Theorem E.9** (Second-order MeanFlow sampling with $T$-step Euler solver belongs to $\mathsf{TC}^0$, formal version of Theorem 4.4). *Given $T \in \mathbb{Z}_+$ as the total iteration of sampling, and arbitrary $\{t_i\}_{i=1}^{T}$ that satisfy $t_i > t_{i+1}$ for all $i \in \{1, \ldots, T - 1\}$ as timestep scheduler. Assume the precision $p \leq \text{poly}(n)$, the number of transformer layers $m = O(1)$, and $T = O(1)$. Then we can use a size bounded by $\text{poly}(n)$ and constant depth $2T(m(17d_{\text{std}} + 8d_{\oplus} + 2d_{\text{sqrt}} + d_{\exp}) + 5d_{\text{std}}) + Td_{\text{std}}$ uniform threshold circuit to simulate the MeanFlow $T$-step sampling defined in Definition 4.2.*

*Proof.* By using the result of Theorem 4.3, we can apply a size bounded by $\mathrm{poly}(n)$ and a constant depth $T(m(17d_{\mathrm{std}} + 8d_{\oplus} + 2d_{\mathrm{sqrt}} + d_{\exp}) + 5d_{\mathrm{std}})$ uniform threshold circuit to simulate the computation of first two terms of Definition 4.2.

Then, for the last term, we can also use the result of Theorem 4.3, except we need to add a scaler multiplication $(t_i - t_{i-1})/2$ before the scalar-matrix multiplication. We can simulate this operation by a uniform threshold circuit with size of $1 < \mathrm{poly}(n)$ and a constant depth of $Td_{\mathrm{std}}$.

Thus, we can have that the size of the uniform threshold circuit is bounded by $\mathrm{poly}(n)$, and the total depth of the circuit is $2T(m(17d_{\mathrm{std}} + 8d_{\oplus} + 2d_{\mathrm{sqrt}} + d_{\exp}) + 5d_{\mathrm{std}}) + Td_{\mathrm{std}} = O(1)$ total depth since $T = O(1)$ and $m = O(1)$. $\square$

## F  PROVABLY EFFICIENT CRITERIA

This section provides the missing proofs from Section 5. In Section F.1, we show the sampling runtime for the original second-order MeanFlow. In Section F.2, we present the sampling runtime for both the fast second-order MeanFlow. In Section F.3, we provide an error bound between the fast second-order MeanFlow layer and the second-order MeanFlow layer.

### F.1  INFERENCE RUNTIME OF SECOND-ORDER MeanFlow

In this section, we present the inference runtime of second-order MeanFlow. The following lemma establishes the time complexity for the inference process of second-order MeanFlow.

**Lemma F.1** (Sampling runtime of Second-order MeanFlow, formal version of Lemma 5.3)**.** *Consider the original second-order* MeanFlow *inference pipeline. The input is a tensor* $\mathsf{X} \in \mathbb{R}^{h \times w \times c}$, *where the height $h$ and width $w$ are both equal to $n$, and the number of channels $c$ is on the order of $O(\log n)$. The interpolated state at time $t \in [0, 1]$ is denoted by $\mathsf{F}^t$, with $\mathsf{F}^1$ representing the final state. The model architecture consists of attention* (Attn)*, MLP* (MLP$(\cdot, c, d)$)*, and Layer Normalization* (LN) *layers.*

*Based on these conditions, the inference time complexity of second-order* MeanFlow *is bounded by* $O(n^{4+o(1)})$.

*Proof.* **Part 1: Runtime of first-order** MeanFlow **layer.** Each first-order MeanFlow layer processes an input of size $n \times n \times c$. The computation is dominated by the attention mechanism, which has a runtime of $O(n^4 c)$ per layer. The total runtime for all first-order layers is the sum over these individual complexities:

$$\mathcal{T}_{\mathsf{MF}} = O(n^{4+o(1)}).$$

where the first step is obtained by simple algebra and $c = O(\log n)$. Here MF denotes MeanFlow.

**Part 2: Runtime of second-order** MeanFlow **layer.** Similarly, each second-order MeanFlow layer operates on an input of size $n \times n \times c$, with the attention mechanism being the computational bottleneck at $O(n^4 c)$ per layer. The runtime complexity for all second-order layers is:

$$\mathcal{T}_{\mathsf{SMF}} = O(n^{4+o(1)}),$$

wherhe the first is obtained by simple algebra and $c = O(\log n)$. Here SMF denotes second-order MeanFlow.

The total runtime for the entire high-order MeanFlow architecture is the sum of the runtimes of the first-order and second-order components.

$$\mathcal{T}_{\mathrm{ori}} = \mathcal{T}_{\mathsf{MF}} + \mathcal{T}_{\mathsf{SMF}} = O(n^{4+o(1)}).$$

Thus, we complete the proof. $\square$

### F.2  INFERENCE RUNTIME OF FAST SECOND-ORDER MeanFlow

In this section, we analyze the inference runtime of fast second-order MeanFlow.

**Lemma F.2** (Sampling runtime of fast Second-order MeanFlow, formal version of Lemma 5.4). *Consider the fast second-order MeanFlow inference pipeline. It takes an input tensor $Z_1 \in \mathbb{R}^{h \times w \times c}$, where the height $h = n$, width $w = n$, and the number of channels $c = O(\log n)$. The model architecture consists of attention (AAttC), MLP (MLP$(\cdot, c, d)$), and Layer Normalization (LN) layers. The interpolated state at time $t \in [0, 1]$ is denoted by $Z_t$, with $Z_0$ as the final state. Based on these conditions, the total inference runtime complexity is bounded by $O(n^{2+o(1)})$.*

*Proof.* **Part 1: Runtime of fast first-order** MeanFlow **layer.** Each fast first-order MeanFlow layer processes an input of size $n \times n \times c$. The layer's runtime is determined by its two main components: the MLP and the attention mechanism. First, the MLP layer requires $O(n^2 c)$ time. Given that the number of channels $c = O(\log n)$, this complexity is $O(n^{2+o(1)})$. Second, leveraging the method from (Alman & Song, 2023), the computationally expensive attention mechanism is accelerated from $O(n^4 c)$ to $O(n^2 c)$, which is also $O(n^{2+o(1)})$.

Since the runtime of both key components is $O(n^{2+o(1)})$, the complexity for a single layer is dominated by this term. Assuming a constant or polylogarithmic number of layers, the total runtime for all fast first-order MeanFlow layers is: $\mathcal{T}_{\mathsf{MF_{fast}}} = O(n^{2+o(1)})$.

**Part 2: Runtime of fast second-order** MeanFlow **layer.** The runtime analysis for the fast second-order MeanFlow layers is similar to that of the first-order layers. Each layer processes an input of size $n \times n \times c$. The key computational components—the MLP layer and the accelerated attention mechanism from (Alman & Song, 2023)—both operate with a complexity of $O(n^{2+o(1)})$.

Therefore, the total runtime complexity for all fast second-order layers is also bounded by this term: $\mathcal{T}_{\mathsf{SMF_{fast}}} = O(n^{2+o(1)})$. Here $\mathsf{SMF_{fast}}$ denotes fast second-order MeanFlow.

Thus, combining Part 1 and Part 2, we obtain the total runtime complexity for the fast high-order MeanFlow, which is

$$\mathcal{T}_{\text{fast}} = \mathcal{T}_{\mathsf{MF_{fast}}} + \mathcal{T}_{\mathsf{SMF_{fast}}} = O(n^{2+o(1)}).$$

Thus, we complete the proof. $\qquad\square$

### F.3 Error Bound between Fast and Standard Second-Order MeanFlow

In this section, we show the error bound between fast and standard second-order MeanFlow. The following lemma quantifies the error introduced by the approximate attention mechanism (Definition 5.1).

**Lemma F.3** (Error bound between fast second-order MeanFlow layer and standard second-order MeanFlow layer, formal version of Lemma 5.7). *For the error analysis of the second-order MeanFlow Layer, we establish the following conditions. Let $Z_1 \in \mathbb{R}^{h \times w \times c}$ be the input and let $Z'_1 \in \mathbb{R}^{h \times w \times c}$ be its approximation, such that the approximation error is bounded by $\|Z_1 - Z'_1\|_\infty \leq \epsilon$ for some small constant $\epsilon > 0$.*

*The analysis considers interpolated inputs $Z_t, Z_{\text{fast},t} \in \mathbb{R}^{h \times w \times c}$ over a time step $t \in [0, 1]$. These inputs are processed by a standard second-order MeanFlow layer, SMF$(\cdot, \cdot, \cdot)$, and a fast variant, SMF$_{\text{fast}}(\cdot, \cdot, \cdot)$, where the fast variant substitute Attn operation with AAttC (Definition 5.1). The approximation is achieved via a polynomial $f$ of degree $g$ and low-rank matrices $U, V \in \mathbb{R}^{hw \times k}$.*

*We assume all matrix entries are bounded by a constant $R > 1$. Crucially, we also make assumption that the LayerNorm function LN$(\cdot)$ (Definition C.31), does not exacerbate error propagation; that is, if $\|Z'_1 - Z_1\|_\infty \leq \epsilon$, then it follows that $\|\mathsf{LN}(Z'_1) - \mathsf{LN}(Z_1)\|_\infty \leq \epsilon$.*

*Then, we have*

$$\|\mathsf{SMF}_{\text{fast}}(Z_{\text{fast},t}, t, r) - \mathsf{SMF}(Z_t, t, r)\|_\infty \leq O(c^2 k R^{g+2}) \cdot \epsilon.$$

*Proof.* We begin by establishing a bound on the difference between the interpolated inputs, $Z_{\text{fast},t}$ and $Z_t$. Given that $t \in [0, 1]$ and the input approximation error is bounded by $\|Z'_1 - Z_1\|_\infty \leq \epsilon$, we have:

$$\|Z_{\text{fast},t} - Z_t\|_\infty = \|t(Z'_1 - Z_1)\|_\infty$$

$$\leq \|Z_1' - Z_1\|_\infty$$
$$\leq \epsilon, \tag{12}$$

where the first step follows from the definition of linear interpolated flows, the second step follows from $t \in [0,1]$, and the second step follows from $\|Z_0' - Z_0\|_\infty \leq \epsilon$.

Then, according to Definition 4.2, the predicted flows are computed using:

$$Z_r = \mathsf{SMF}(Z_t, t, r), \tag{13}$$
$$Z_{\mathrm{fast},r} = \mathsf{SMF}_{\mathrm{fast}}(Z_{\mathrm{fast},t}, t, r). \tag{14}$$

Let $||$ denote the concatenation operation. Next, we bound the error for two different outputs:

$$\|Z_r - Z_{\mathrm{fast},r}\|_\infty = \|\mathsf{SMF}(Z_t, t, r) - \mathsf{SMF}_{\mathrm{fast}}(Z_{\mathrm{fast},t}, t, r)\|_\infty$$

$$= \|Z_t + (r-t)\mathsf{ViT}_1(Z_t \,\|\, r\mathbf{1}_n \,\|\, t\mathbf{1}_n)_{*,1:d} + \frac{1}{2}(r-t)^2\mathsf{ViT}_2(Z_t \,\|\, r\mathbf{1}_n \,\|\, t\mathbf{1}_n)_{*,1:d} - Z_{\mathrm{fast},t}$$

$$- (r-t)\mathsf{ViT}_{\mathrm{fast},1}(Z_{\mathrm{fast},t} \,\|\, r\mathbf{1}_n \,\|\, t\mathbf{1}_n)_{*,1:d} - \frac{1}{2}(r-t)^2\mathsf{ViT}_{\mathrm{fast},2}(Z_{\mathrm{fast},t} \,\|\, r\mathbf{1}_n \,\|\, t\mathbf{1}_n)_{*,1:d}\|_\infty$$

$$\leq \|Z_t - Z_{\mathrm{fast},t}\|_\infty + (r-t)\|(\mathsf{ViT}_1(Z_t \,\|\, r\mathbf{1}_n \,\|\, t\mathbf{1}_n) - \mathsf{ViT}_{\mathrm{fast},1}(Z_{\mathrm{fast},t} \,\|\, r\mathbf{1}_n \,\|\, t\mathbf{1}_n))_{*,1:d}\|_\infty$$

$$+ \frac{1}{2}(r-t)^2\|(\mathsf{ViT}_2(Z_t \,\|\, r\mathbf{1}_n \,\|\, t\mathbf{1}_n) - \mathsf{ViT}_{\mathrm{fast},2}(Z_{\mathrm{fast},t} \,\|\, r\mathbf{1}_n \,\|\, t\mathbf{1}_n))_{*,1:d}\|_\infty$$

$$\leq \epsilon + A + B,$$

where the first step follows from substituting $Z_r$ and $Z_{\mathrm{fast},r}$ with Eq. (13) and Eq. (14), the second step follows from Definition 4.2, the third step follows from triangle inequality, the last step follows from Eq. (12) and defining two shorthand notations:

$$A := (r-t)\|(\mathsf{ViT}_1(Z_t \,\|\, r\mathbf{1}_n \,\|\, t\mathbf{1}_n) - \mathsf{ViT}_{\mathrm{fast},1}(Z_{\mathrm{fast},t} \,\|\, r\mathbf{1}_n \,\|\, t\mathbf{1}_n))_{*,1:d}\|_\infty,$$

$$B := \frac{1}{2}(r-t)^2\|(\mathsf{ViT}_2(Z_t \,\|\, r\mathbf{1}_n \,\|\, t\mathbf{1}_n) - \mathsf{ViT}_{\mathrm{fast},2}(Z_{\mathrm{fast},t} \,\|\, r\mathbf{1}_n \,\|\, t\mathbf{1}_n))_{*,1:d}\|_\infty.$$

Consider the $Y$ in the first layer of ViT,

$$\|Y_1 - Y_{\mathrm{fast},1}\|_\infty = \|\mathsf{Attn}(\mathsf{LN}_{1,1}(Z_t \,\|\, r\mathbf{1}_n \,\|\, t\mathbf{1}_n)) + Z_t - \mathsf{AAttC}(\mathsf{LN}_{1,1}(Z_{\mathrm{fast},t} \,\|\, r\mathbf{1}_n \,\|\, t\mathbf{1}_n)) - Z_{\mathrm{fast},t}\|_\infty$$

$$\leq \|\mathsf{Attn}(\mathsf{LN}_{1,1}(Z_t \,\|\, r\mathbf{1}_n \,\|\, t\mathbf{1}_n)) - \mathsf{AAttC}(\mathsf{LN}_{1,1}(Z_{\mathrm{fast},t} \,\|\, r\mathbf{1}_n \,\|\, t\mathbf{1}_n))\|_\infty + \|Z_t - Z_{\mathrm{fast},t}\|_\infty$$

$$\leq O(kR^{g+1}c) \cdot \epsilon + \epsilon, \tag{15}$$

where the first step is due to Definition C.32, the second step is obtained by triangular inequality, and the last step follows from Lemma 5.5 and assumption in the lemma.

Then, for $X$ in the first layer of ViT,

$$\|X_1 - X_{\mathrm{fast},1}\|_\infty = \|\mathsf{MLP}_1(\mathsf{LN}_{1,2}(Y_1)) + Y_1 - \mathsf{MLP}_1(\mathsf{LN}_{1,2}(Y_{\mathrm{fast},1})) - Y_{\mathrm{fast},1}\|_\infty$$

$$\leq \|\mathsf{MLP}_1(\mathsf{LN}_{1,2}(Y_1)) - \mathsf{MLP}_1(\mathsf{LN}_{1,2}(Y_{\mathrm{fast},1}))\|_\infty + \|Y_1 - Y_{\mathrm{fast},1}\|_\infty$$

$$\leq cR(O(kR^{g+1}c) \cdot \epsilon + \epsilon) + O(kR^{g+1}c) \cdot \epsilon + \epsilon$$

$$\leq O(c^2kR^{g+2}) \cdot \epsilon,$$

where the first step follows from Definition C.32, the second step is due to triangular inequality, the third step is obtained by Lemma 5.6, Eq. (15), and assumption in the lemma, and the last step follows from simple algebra.

Thus we have

$$A \leq (r-t)O(c^2kR^{g+2}) \cdot \epsilon = O(c^2kR^{g+2}) \cdot \epsilon,$$

$$B \leq \frac{1}{2}(r-t)^2O(c^2kR^{g+2}) \cdot \epsilon = O(c^2kR^{g+2}) \cdot \epsilon.$$

Thus we can combine the final result,

$$\|Z_r - Z_{\mathrm{fast},r}\|_\infty \leq \epsilon + A + B \leq O(c^2kR^{g+2}) \cdot \epsilon.$$

Thus we complete the proof. $\qquad\square$

## LLM USAGE DISCLOSURE

LLMs were used only to polish language, such as grammar and wording. These models did not contribute to idea creation or writing, and the authors take full responsibility for this paper's content.

