# OpenReview forum: "Towards High-Order Mean Flow Generative Models: Feasibility, Expressivity, and Provably Efficient Criteria"
_ICLR.cc/2026/Conference — Submitted to ICLR 2026_

### Official Review · Reviewer_169t · 2025-10-28

**Soundness:** 3
**Presentation:** 2
**Contribution:** 3
**Rating:** 6
**Confidence:** 1

**Summary:**

This paper introduces Second-Order MeanFlow, a theoretical extension of the MeanFlow framework that incorporates average acceleration fields alongside average velocities for generative modeling. The authors provide a rigorous mathematical treatment demonstrating that the proposed formulation satisfies a generalized consistency condition enabling stable one-step sampling, possesses expressivity within the TC⁰ circuit complexity class, and admits an efficient implementation using approximate attention with provable error bounds and nearly quadratic time complexity. By grounding these results in detailed proofs of feasibility, expressivity, and computational efficiency, the work establishes a strong theoretical foundation for exploring higher-order dynamics in simulation-free generative models and points toward potential future applications in fast, expressive generative modeling.

**Strengths:**

1. The paper provides formal definitions, clear assumptions, and detailed proofs for all major results. Each theorem (on feasibility, expressivity, and efficiency) is logically consistent and well supported by prior work in flow matching and circuit complexity.

2. Extending MeanFlow to second-order dynamics is a meaningful and nontrivial contribution. By modeling average accelerations, SOMF potentially captures richer temporal dependencies in generative flows, offering a principled direction toward high-order simulation-free models.

3. The circuit complexity analysis situates SOMF within the TC⁰ class, establishing theoretical efficiency guarantees and connecting generative modeling to computational complexity.

**Weaknesses:**

1. The paper remains entirely theoretical, and while the authors acknowledge this, the absence of even small-scale experiments or numerical simulations limits the understanding of how SOMF behaves in practice. Demonstrating a toy example or a simple empirical comparison would significantly enhance credibility.

2. The motivation for introducing acceleration terms is described formally but not intuitively. The paper could benefit from geometric or dynamical explanations illustrating how incorporating second-order information improves expressivity or sampling quality.

**Questions:**

1. While the theoretical results are strong, the real-world benefits of Second-Order MeanFlow remain unclear. How might this framework impact model expressivity and model performance in practice?

2. Although the proofs are rigorous, the paper’s density might limit accessibility to a broader audience. The authors could consider including intuitive figures (e.g., geometric interpretation of first- vs second-order flows) or summary tables that condense key theorems and assumptions, helping readers grasp the main ideas without diving into all formal details.

---

> ### Author Response · Authors · 2025-11-23
>
> We sincerely thank the reviewer for the thoughtful feedback and for recognizing the soundness, clarity of proofs, and theoretical significance of our work. Below, we address the raised weaknesses and questions in detail.
>
> ## Weakness 1 & Question 1: Lack of experiment
>
> Thank you for your question regarding empirical validation. While we agree that experimental results can provide additional insight, the main goal of this paper is to establish theoretical foundations. Validating our theoretical guarantees empirically would require substantial implementation effort and engineering that fall outside the current scope. Our primary contribution lies in the theoretical analysis, including tight upper and lower bounds. Such purely theoretical works are common in top machine learning venues (e.g., NeurIPS, ICLR, ICML). For example, ICLR has accepted papers such as [3,4,5,6] that focus exclusively on theory without empirical evaluation. Similar examples exist at ICML [7,8,9] and NeurIPS [1,2]. Our work follows this line, particularly in the study of diffusion generative models [2,3,4], and we believe it contributes meaningfully to the theoretical understanding of these models.
>
> ## Weakness 2: Insufficient intuitive motivation
>
> We appreciate this suggestion. We will enrich the introduction with geometric intuition. In brief, first-order MeanFlow models an average velocity field that approximates the path between distributions, but it cannot directly encode curvature in the probability flow. The second-order extension introduces average acceleration, which allows the model to account for trajectory bending and temporal momentum.
>
> Geometrically, this means SOMF captures not only where the flow moves but also how its direction changes. This additional degree of freedom improves expressivity without requiring iterative integration, enabling more accurate single-step sampling. We will clarify this intuition in the revised version with a simple diagram comparing first- and second-order trajectories.
>
>
> ## Question 2: Accessibility and presentation improvements
>
> We appreciate this feedback and will improve presentation clarity. In a future revision, we will:
>
> - Add Figure 1 illustrating geometric intuition of first- vs. second-order flows;
>
> - Provide a summary table listing all key theorems, assumptions, and complexity results;
>
> - Include a high-level algorithm box summarizing SOMF training and inference;
>
> - Move several technical proofs to the appendix for better readability.
>
> ### References
>
>
> [1] Damian, Alex, Jason D. Lee, and Joan Bruna. The Generative Leap: Tight Sample Complexity for Efficiently Learning Gaussian Multi-Index Models. NeurIPS 2025.
>
> [2] Hu, Jerry Yao-Chieh, Weimin Wu, Zhuoru Li, Sophia Pi, Zhao Song, and Han Liu. On Statistical Rates and Provably Efficient Criteria of Latent Diffusion Transformers (DiTs). NeurIPS 2024.
>
> [3] Benton, Joe, Valentin De Bortoli, Arnaud Doucet, and George Deligiannidis. Nearly d-linear convergence bounds for diffusion models via stochastic localization. ICLR 2024.
>
> [4] Yao-Chieh Hu, Jerry, Weimin Wu, Yi-Chen Lee, Yu-Chao Huang, Minshuo Chen, and Han Liu. On Statistical Rates of Conditional Diffusion Transformers: Approximation, Estimation and Minimax Optimality. ICLR 2025.
>
> [5] Furuya, Takashi, Koichi Taniguchi, and Satoshi Okuda. Quantitative Approximation for Neural Operators in Nonlinear Parabolic Equations. ICLR 2025.
>
> [6] Kim, Juno, Dimitri Meunier, Arthur Gretton, Taiji Suzuki, and Zhu Li. Optimality and adaptivity of deep neural features for instrumental variable regression. ICLR 2025.
>
> [7] Han Zhong, Jiachen Hu, Yecheng Xue, Tongyang Li, and Liwei Wang. Provably Efficient Exploration in Quantum Reinforcement Learning with Logarithmic Worst-Case Regret. ICML 2024.
>
> [8] Bo Xue, Dake Bu, Ji Cheng, Yuanyu Wan, and Qingfu Zhang. Multi-objective Linear Reinforcement Learning with Lexicographic Rewards. ICML 2025.
>
> [9] Lambrechts, Gaspard, Damien Ernst, and Aditya Mahajan. A Theoretical Justification for Asymmetric Actor-Critic Algorithms. ICML 2025.

---

### Official Review · Reviewer_ds4t · 2025-10-30

**Soundness:** 4
**Presentation:** 2
**Contribution:** 3
**Rating:** 4
**Confidence:** 3

**Summary:**

The paper “Towards High-Order Mean Flow Generative Models: Feasibility, Expressivity, and Provably Efficient Criteria” presents a theoretical analysis of Second-Order MeanFlow, an extension of the MeanFlow framework for generative modeling. Whereas MeanFlow replaces instantaneous velocity fields with average velocities to allow efficient single-step sampling, Second-Order MeanFlow further introduces average acceleration fields to model higher-order dynamics. The authors show that this formulation satisfies a generalized consistency condition enabling stable one-step inference, prove that the model’s expressivity lies within the TC⁰ circuit complexity class, and demonstrate provable efficiency through approximate attention computations with O(n^{2+o(1)}) time complexity and 1/poly(n) approximation error. The study provides theoretical support for the feasibility and tractability of high-order extensions to MeanFlow models.

**Strengths:**

- The paper is original in that it bridges the fields of generative modeling and circuit complexity. While several works on transformers leverage circuit complexity techniques, these haven’t been applied in the context of generative modeling.

- In particular, the finding that MeanFlows and high order MeanFlows are in TC0, which is the same class to which transformers without CoT belong, is very interesting, and may prompt researchers to think of CoT-like approaches for flows.

**Weaknesses:**

- The main weakness of the paper is that it does not include experiments testing the performance of the second-order MeanFlow. This may not be perceived as a weakness in a theoretical computer science venue, but ICLR is a general machine learning conference and the empirical performance of proposed algorithm must be shown. In that sense, this work is incomplete, as we do not know whether second-order MeanFlow is only a theoretical construction, or whether it has applications in practice.

- Small comment: The meaning of ViT (Vision Transformers) needs to be clarified in the first appearance of the expression.

- Section B in the Appendix is illuminating and explains the purpose of the paper, which is not clear from the main text: the reason for higher order MeanFlow and for proving circuit complexity are only briefly discussed in the introduction, but not in detail. The explanation in Section B should appear in the main text, and arguably in the first part of the paper. In particular the practical implications are too important to the story of the paper to be left in the appendix.

- The writing style is very dry. The main text is essentially a list of definitions and results. The authors should present a subset of their results in the main text, and defer the other ones to the appendix, and move the content of Section B to the main text.

**Questions:**

- Is there a reason the authors decided to prove circuit complexity results on MeanFlow and higher order MeanFlow instead of on standard flow matching or diffusion models? What I mean is, it looks like this paper has two topics: high order MeanFlow, and circuit complexity techniques for generative modeling. Is there a reason this is one paper instead of two?

- Would the authors be able to include experimental results on higher order MeanFlow? I would be willing to update my score if they provide compelling experiments.

---

> ### Author Response · Authors · 2025-11-23
>
> We thank the reviewer for their careful reading and constructive feedback. Below, we address all identified weaknesses and questions in detail.
>
> ## Weakness 1 & Question 2: Lack of experiments
>
> Thank you for your question regarding empirical validation. While we agree that experimental results can provide additional insight, the main goal of this paper is to establish theoretical foundations. Validating our theoretical guarantees empirically would require substantial implementation effort and engineering that fall outside the current scope. Our primary contribution lies in the theoretical analysis, including tight upper and lower bounds. Such purely theoretical works are common in top machine learning venues (e.g., NeurIPS, ICLR, ICML). For example, ICLR has accepted papers such as [3,4,5,6] that focus exclusively on theory without empirical evaluation. Similar examples exist at ICML [7,8,9] and NeurIPS [1,2]. Our work follows this line, particularly in the study of diffusion generative models [2,3,4], and we believe it contributes meaningfully to the theoretical understanding of these models.
>
> ## Weakness 2: Clarification of “ViT”
>
> We thank the reviewer for catching this. We indeed use ViT to denote “Vision Transformer,” which serves as a concrete instantiation of the sampling network in Definitions 4.1–4.2. We will clarify its first appearance.
>
> ##  Weakness 3: Appendix Section B should be moved to main text
> We completely agree. Section B (“Discussion”) indeed provides essential context—explaining why higher-order MeanFlow is introduced and why circuit complexity matters for generative modeling. In the revision, we will move this discussion into Section 1 (Introduction) and condense the main motivations.
> We thank the reviewer for highlighting that this connection deserves emphasis in the main body.
> ## Weakness 4: Writing and structure
>
> We appreciate this suggestion and will make the text more engaging and accessible. Specifically:
>
> - We will move some technical proofs (currently in Sections 3–5) to the Appendix.
>
> - We will add geometric intuition figures comparing first- and second-order MeanFlows (depicting curvature in trajectories).
>
> These adjustments will make the paper more readable while maintaining mathematical rigor.
>
> ## Question 1: Why MeanFlow for Circuit Complexity (One paper vs. Two)?
>
> 1. Why MeanFlow vs. Standard Diffusion/FM?
>
> Standard Diffusion or Flow Matching models typically solve ODEs requiring many steps ($T$) for high quality. This sequential dependence increases circuit depth with $T$, making it difficult to bound within the constant-depth $TC^0$ class without strong assumptions.
>
> In contrast, MeanFlow uses average velocity (and acceleration) satisfying a generalized consistency condition, enabling efficient single-step sampling. This allows treating $T$ as $O(1)$ (Theorems 4.3, 4.4), the key prerequisite for proving the sampling process belongs to $TC^0$.
>
> 2. Why One Paper?
>
> The paper aims to establish a "full stack" of theoretical guarantees for this new high-order architecture:
>
> Feasibility: Can we define it consistent? (Yes).
>
> Expressivity: Is it computationally bounded? (Yes, $TC^0$).
>
> Efficiency: Can we compute it fast? (Yes, almost quadratic time).
>
> ### References
>
> [1] Damian, Alex, Jason D. Lee, and Joan Bruna. The Generative Leap: Tight Sample Complexity for Efficiently Learning Gaussian Multi-Index Models. NeurIPS 2025.
>
> [2] Hu, Jerry Yao-Chieh, Weimin Wu, Zhuoru Li, Sophia Pi, Zhao Song, and Han Liu. On Statistical Rates and Provably Efficient Criteria of Latent Diffusion Transformers (DiTs). NeurIPS 2024.
>
> [3] Benton, Joe, Valentin De Bortoli, Arnaud Doucet, and George Deligiannidis. Nearly d-linear convergence bounds for diffusion models via stochastic localization. ICLR 2024.
>
> [4] Yao-Chieh Hu, Jerry, Weimin Wu, Yi-Chen Lee, Yu-Chao Huang, Minshuo Chen, and Han Liu. On Statistical Rates of Conditional Diffusion Transformers: Approximation, Estimation and Minimax Optimality. ICLR 2025.
>
> [5] Furuya, Takashi, Koichi Taniguchi, and Satoshi Okuda. Quantitative Approximation for Neural Operators in Nonlinear Parabolic Equations. ICLR 2025.
>
> [6] Kim, Juno, Dimitri Meunier, Arthur Gretton, Taiji Suzuki, and Zhu Li. Optimality and adaptivity of deep neural features for instrumental variable regression. ICLR 2025.
>
> [7] Han Zhong, Jiachen Hu, Yecheng Xue, Tongyang Li, and Liwei Wang. Provably Efficient Exploration in Quantum Reinforcement Learning with Logarithmic Worst-Case Regret. ICML 2024.
>
> [8] Bo Xue, Dake Bu, Ji Cheng, Yuanyu Wan, and Qingfu Zhang. Multi-objective Linear Reinforcement Learning with Lexicographic Rewards. ICML 2025.
>
> [9] Lambrechts, Gaspard, Damien Ernst, and Aditya Mahajan. A Theoretical Justification for Asymmetric Actor-Critic Algorithms. ICML 2025.

---

### Official Review · Reviewer_HNp8 · 2025-11-01

**Soundness:** 2
**Presentation:** 1
**Contribution:** 2
**Rating:** 2
**Confidence:** 3

**Summary:**

The paper extends the MeanFlow framework from first-order (average velocity) to second-order (average acceleration). It demonstrates that the new average-acceleration field satisfies a consistency condition, enabling single-step or few-step sampling. It shows that the resulting samplers, when implemented with a constant-depth ViT and a fixed number of Euler steps, lie in the circuit class TC0. It demonstrates an almost-quadratic-time implementation by replacing exact attention with a fast approximate version, while bounding the induced error.

**Strengths:**

1. Clear formalization. The manuscript is meticulous in its definitions, maintaining consistency with prior Flow-Matching literature.
2. The assumptions are explicitly stated.

**Weaknesses:**

1. The main technical ideas are not very novel. Most of the argument builds upon earlier work on the circuit complexity of Transformer or VAR models, such as [1] and [2], and the extension to MeanFlow seems relatively straightforward.

2. There are no empirical or simulated results to illustrate whether the proposed approximations actually improve runtime.

3. The paper relies on many assumptions which may not hold in reality or have little practical guidance. The TC^0 result relies on (i) O(1) Euler steps, (ii) O(1) ViT layers, and (iii) poly-bounded weight magnitudes. In practice, state-of-the-art generative ViTs employ dozens of layers, adaptive samplers, and numerous engineering tricks, so the theorem sheds limited light on real-world deployments. It becomes more questionable when the paper fails to provide any empirical evidence.

4. The scope of the analysis is rather narrow, focusing on the MeanFlow backbone.

5. The connection between the circuit-theoretic results and practical model design remains abstract. It would be better if the authors discuss how the results of the paper could lead to practical guidance on architectural design and training.

References
[1] Chen, Bo, et al. "Circuit Complexity Bounds for RoPE-based Transformer Architecture." arXiv preprint arXiv:2411.07602 (2024).
[2] Li, Xiaoyu, et al. "Theoretical constraints on the expressive power of rope-based tensor attention transformers." CoRR (2024).

**Questions:**

1. I feel that the current analysis is quite ad-hoc. What is the most general architecture that the same analysis could extend to?
2. I am curious if the authors could design some toy experiments for empirical evidence.

---

> ### Author Response · Authors · 2025-11-23
>
> We thank the reviewer for the detailed evaluation and for highlighting the strengths in formalization and clarity of assumptions. Below we address all weaknesses and questions.
>
> ## Weakness 1: The technical ideas are not very novel
>
> While we leverage analysis techniques from [1,2], our contributions are distinct in three key dimensions:
>
> Unlike [1,2] which focus on static sequence-to-sequence Transformers (LLMs), we analyze generative flows. We specifically characterize the complexity of time-dependent velocity and acceleration fields, which is structurally different from standard attention.
>
> The generalized consistency condition for acceleration is a new theoretical contribution. It addresses the unique time-averaged properties of MeanFlow, requiring independent derivations that do not follow from [1,2].
>
> We provide the first $TC^0$ characterization of higher-order generative dynamics. This extends circuit complexity analysis into the previously unexplored domain of flow matching expressivity.
>
> ## Weakness 2 & Question 2: No empirical results.
>
> We appreciate the reviewer raising this point, which was also noted by Reviewer ds4t. As detailed in our response to Reviewer ds4t (please see "Weakness 1 & Question 2: Lack of experiments")
>
> ## Weakness 3: Assumptions may not hold
>
> We clarify that these are standard complexity theory assumptions that map directly to practice:
>
> $O(1)$ Depth: "Constant depth" implies depth is independent of input sequence length $n$. Practical ViTs use a fixed number of layers (e.g., 12 or 24) regardless of input resolution (e.g., $256^2$ vs $512^2$). Thus, real-world ViTs strictly satisfy the $TC^0$ constant-depth condition relative to $n$.
>
> Bounded Weights: The assumption of polynomially bounded weights is consistent with practical implementations using float16/bfloat16 precision and weight decay/normalization (LayerNorm) techniques, which prevent weights from exploding to infinity.
>
> $O(1)$ Euler Steps: Since MeanFlow is explicitly designed for "One-Step" or "Few-Step" generation, assuming $T=O(1)$ is not a simplification but an accurate reflection of the model's intended deployment scenario.
>
> ## Weakness 4 & Question 1: Scope is narrow—focused only on MeanFlow
>
> Our focus on MeanFlow is a deliberate choice to address a distinct paradigm shift in generative modeling.  Unlike standard Flow Matching, MeanFlow models average velocities, uniquely enabling efficient single-step sampling.  However, extending this to high-order dynamics is non-trivial and presents theoretical challenges absent in standard literature.  Our work overcomes the fundamental hurdle of proving that a novel Average Acceleration field satisfies the generalized consistency condition, a necessary step to enable high-order dynamics in this class of fast-inference models.
>
> ## Weakness 5: Connection between circuit theory and practical design is abstract
>
> The connection to practical design is supported by LLM evidence confirming that higher complexity classes are prerequisites for reasoning. For instance, [3] proves that while Transformers are $TC^0$, Chain-of-Thought increases depth to solve $NC^1$-hard problems. Similarly, recurrence shifts models to sequential P, enabling recursive reasoning [4, 5] and graph algorithms [6]. Even positional encoding changes [7] correlate with expressivity.
>
> In this context, our characterization of MeanFlow within $TC^0$ provides architectural guidance rather than abstract theory.  It delineates the exact computational limits of constant-depth, simulation-free flows. This informs practitioners that for tasks requiring complex logical inference (beyond the scope of $TC^0$), scaling parameters is insufficient; instead, one must adopt architectures with greater computational depth, such as recurrent or looped designs, to break the theoretical ceiling.
>
> ### References
>
> [1] Chen, Bo, et al. "Circuit Complexity Bounds for RoPE-based Transformer Architecture." arXiv preprint arXiv:2411.07602 (2024).
>
> [2] Li, Xiaoyu, et al. "Theoretical constraints on the expressive power of rope-based tensor attention transformers." CoRR (2024).
>
> [3] Zhiyuan Li, et al. Chain of Thought Empowers Transformers to Solve Inherently Serial Problems. ICLR 2024.
>
> [4] Angeliki Giannou, et al. Looped Transformers as Programmable Computers. ICML 2023.
>
> [5] Nikunj Saunshi, et al. Reasoning with Latent Thoughts: On the Power of Looped Transformers. ICLR 2025.
>
> [6] Artur Back de Luca, Kimon Fountoulakis. Simulation of Graph Algorithms with Looped Transformers. ICML 2024.
>
> [7] Yang, Songlin, et al. "PaTH Attention: Position Encoding via Accumulating Householder Transformations." arXiv preprint arXiv:2505.16381.

---

### Meta-Review · Area_Chair_v3jq · 2026-01-02

**Summary:**

The reviewers raised several concerns that informed my decision. The most significant and unanimous concern was the complete absence of empirical validation: all three reviewers noted this weakness. Additional concerns included questions about novelty relative to prior circuit complexity work on Transformers (Reviewer HNp8), whether the theoretical assumptions reflect practical deployments (Reviewer HNp8), presentation issues with important motivational content relegated to the appendix (Reviewers ds4t, 169t), and the abstract connection between theoretical results and practical model design.

**Reviewer Concerns:**

- The authors provided reasonable clarification that the assumptions (constant depth, bounded weights, O(1) steps) do align with practical implementations when properly interpreted relative to sequence length
- The explanation of why MeanFlow was chosen over standard diffusion/FM (single-step sampling enables constant-depth analysis) was convincing
- Authors agreed to move Appendix B content to the main text and improve accessibility


- Lack of experiments: This remains the critical unresolved issue. While authors cited precedents of theory-only papers at ML venues, they did not provide any empirical validation. Reviewer ds4t explicitly stated willingness to raise their score given compelling experiments, which were not provided.
- Novelty: Reviewer HNp8's concern that technical ideas build substantially on prior circuit complexity work was only partially addressed
- Practical relevance: The connection between TC0 characterization and actionable architectural guidance remains somewhat abstract despite the rebuttal's discussion of CoT and LLM complexity.

**Reviewer Scores:**

- Reviewer HNp8 (Rating: 2): Would likely remain at 2 or increase marginally to 3. Their core concerns about novelty, practical assumptions, and lack of experiments were not fully resolved. The rebuttal provided clarifications but no new evidence.
- Reviewer ds4t (Rating: 4): Would likely remain at 4. They explicitly conditioned a score increase on "compelling experiments," which were not provided. Their other concerns about presentation were acknowledged but not yet implemented.
- Reviewer 169t (Rating: 6): Would likely remain at 6 or decrease slightly to 5. While they were the most positive reviewer, they also noted the lack of experiments as a weakness, and their confidence level of 1 indicates significant uncertainty in their assessment.

---

### Decision · Program_Chairs · 2026-01-26

Reject